# LangDAug: Langevin Data Augmentation for Multi-Source Domain Generalization in Medical Image Segmentation

Piyush Tiwary [1]  Kinjawl Bhattacharyya [1]  Prathosh A.P. [1]

## Abstract

Medical image segmentation models often struggle to generalize across different domains due to various reasons. Domain Generalization (DG) methods overcome this either through representation learning or data augmentation (DAug). While representation learning methods seek domain-invariant features, they often rely on ad-hoc techniques and lack formal guarantees. DAug methods, which enrich model representations through synthetic samples, have shown comparable or superior performance to representation learning approaches. We propose LangDAug, a novel **Lang**evin **D**ata **Aug**mentation for multi-source domain generalization in 2D medical image segmentation. LangDAug leverages Energy-Based Models (EBMs) trained via contrastive divergence to traverse between source domains, generating intermediate samples through Langevin dynamics. Theoretical analysis shows that LangDAug induces a regularization effect, and for GLMs, it upper-bounds the Rademacher complexity by the intrinsic dimensionality of the data manifold. Through extensive experiments on Fundus segmentation and 2D MRI prostate segmentation benchmarks, we show that LangDAug outperforms state-of-the-art domain generalization methods and effectively complements existing domain-randomization approaches. The codebase for our method is available at https://github.com/backpropagator/LangDAug.

## 1. Introduction

Recent years have seen remarkable advancements in the field of Medical image segmentation, particularly through supervised learning approaches leveraging Convolutional Neural Networks (CNNs) (Milletari et al., 2016; Falk et al., 2019; Zhou et al., 2019; Jin et al., 2019; Isensee et al., 2020; Shen et al., 2017; Olabarriaga & Smeulders, 2001; Asgari Taghanaki et al., 2021; Tiwary et al., 2023). However, a persistent challenge remains: these models often struggle to generalize from the training set (source domain) to unseen test sets (target domain) (Ganin et al., 2016; Kamnitsas et al., 2017; Li et al., 2018b).

The generalization problem stems from the limitations of the Empirical Risk Minimization (ERM) framework, which often performs poorly on out-of-distribution (OOD) test or target domains (Shalev-Shwartz & Ben-David, 2014; Murphy, 2022; Hart et al., 2000; Hajek & Raginsky, 2021). This is primarily due to the i.i.d. assumption underlying ERM, where models minimize the loss solely with respect to the training data distribution. As a result, ERM-trained models exhibit degraded performance on test data from different distributions, violating the i.i.d. assumption and leading to sub-optimal generalization to unseen domains (Vapnik, 1999).

The challenge is particularly pronounced in medical imaging, where domain shifts significantly impact model performance. These shifts arise from factors such as differences in imaging protocols across institutions, variations in equipment specifications, diverse patient attributes (e.g., age, gender, demographics), and disparities between modalities (e.g., CT, MRI, Ultrasound). As noted by Guan & Liu (2021), such shifts hinder the deployment of machine learning models in new clinical settings. The discrepancies between training and target domains can result in inconsistent performance, potentially compromising the model's clinical utility and patient care. Addressing these issues is critical for developing robust and generalizable models.

Various approaches have been proposed to address domain generalization. A key strategy is representation learning, which seeks to learn either domain-invariant (DI) features or sufficiently diverse features to generalize to unseen domains. These methods often disentangle domain-invariant and domain-specific (DS) features, with some approaches discarding domain-specific features entirely and others combining DI-DS pairs to create richer features. Another ap-

---

[1]Department of Electrical Communication Engineering, Indian Institute of Science, Bangalore, India. Correspondence to: Piyush Tiwary <piyushtiwary@iisc.ac.in>.

*Proceedings of the 42nd International Conference on Machine Learning*, Vancouver, Canada. PMLR 267, 2025. Copyright 2025 by the author(s).

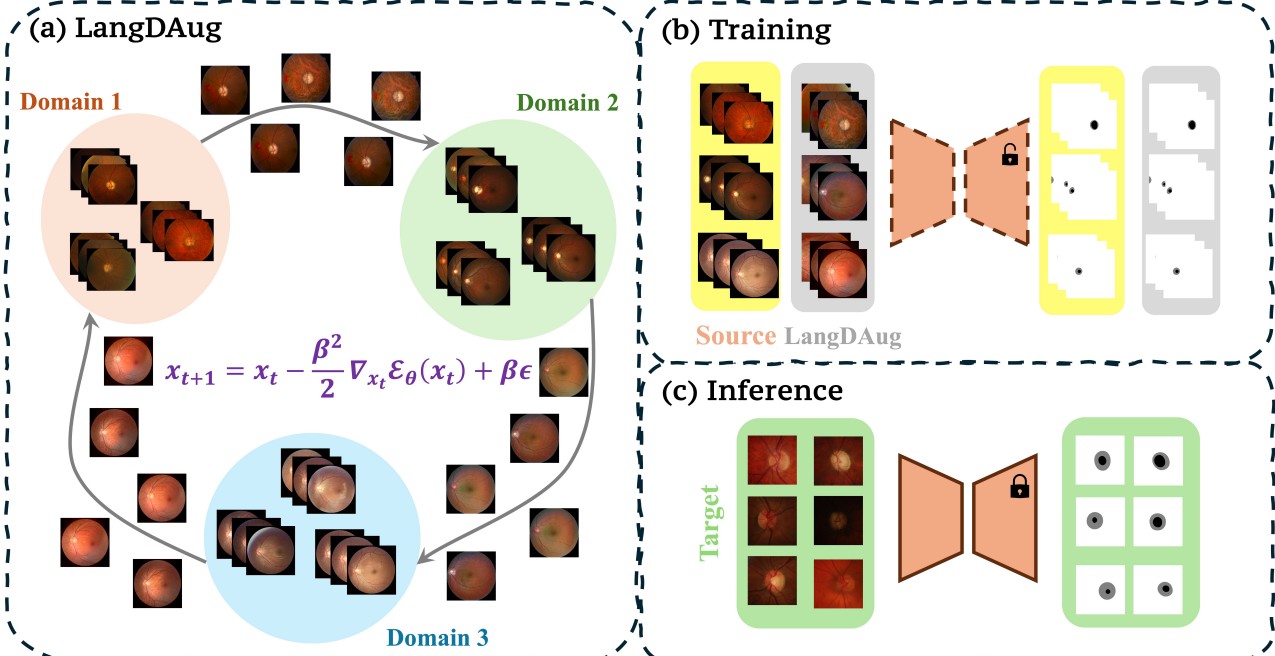

Figure 1: Overview of the proposed method: (a) We run Langevin dynamics (LD) using trained EBMs to transverse between different domains. The intermediate iterates of LD (called Langevin samples) are stored and used for augmentation, (b) The segmentation model is trained with original domain samples as well as Langevin samples, (c) The trained model is then deployed on unseen target domains.

proach is data augmentation, which enhances model representations by supplementing original samples with augmented ones. As shown by (Gulrajani & Lopez-Paz, 2021), data augmentation can achieve comparable performance to representation learning while being simpler to implement.

Our work adopts the data augmentation approach to improve multi-source domain generalization in 2D medical image segmentation. We propose a novel **Lang**evin **D**ata **Aug**mentation (**LangDAug**) scheme that leverages Langevin dynamics (LD) iterates to augment the original dataset. Specifically, we train Energy-Based Models (EBMs) to traverse between different source domains using a contrastive-divergence (CD) objective. This traversal is achieved via iterative gradient-based MCMC methods like LD, with the iterates representing samples from the manifold bridging the source domains. These samples are then incorporated into the ERM framework for downstream task training, potentially enhancing the model's ability to generalize to unseen domains. Our key contributions are:

1. We introduce LangDAug, an augmentation technique for multi-source domain generalization in 2D medical image segmentation. By training EBMs for each pair of source domains via CD, we generate intermediate domain samples that augment the source domains within the ERM framework.

2. We provide theoretical insights into LangDAug's effects, demonstrating its regularization impact on ERM training. This leads to Hessian-smoothing and flatter minima, aiding generalization to unseen domains.

3. We conduct rigorous evaluations on two medical image segmentation benchmarks: fundus segmentation and 2D MRI image segmentation. Our results show that LangDAug outperforms state-of-the-art domain generalization methods and can enhance the performance of domain-randomization based generalization approaches.

## 2. Related Works

### 2.1. Domain Generalization

Domain Generalization (DG) frameworks (Motiian et al., 2017; Wang et al., 2022; Zhou et al., 2022a; Dou et al., 2019; Gu et al., 2021; Cong et al., 2022) aim to train a single model using one or multiple source domains to achieve robust performance on unseen target domains. These methods typically focus on either learning domain-invariant features or developing diverse features capable of generalizing to target domains. Meta-learning DG methods (Li et al., 2018a; Liu et al., 2021b;a; 2020; Finn et al., 2017; Balaji et al., 2018; Du et al., 2020b; Tiwary et al., 2024; 2025) simulate

domain shifts by splitting source data into meta-train/test sets.

Current state-of-the-art DG approaches can be broadly categorized into two main types (Wang et al., 2022): representation learning-based methods and data manipulation-based methods. Representation learning methods (Li et al., 2018c; Muandet et al., 2013; Upchurch et al., 2017; Gardner et al., 2015; Zhang et al., 2022; Li et al., 2022b; 2021b; Zhou et al., 2021; Wang et al., 2020; Chen et al., 2023; Pan et al., 2018; Hu et al., 2022) are further subdivided into domain-invariant (DI) representation learning and feature disentanglement approaches. Domain-invariant methods (Blanchard et al., 2021; Hu et al., 2020; Li et al., 2018b; Garg et al., 2021; Pan et al., 2018; Arjovsky et al., 2019) minimize cross-domain feature discrepancies—e.g., Fish aligns inter-domain gradients, Fishr regularizes gradient variances, and Hutchinson matches Hessian matrices. While effective, these may sacrifice domain-specific (DS) features to balance single- and multi-domain performance. Another line of work uses test time adaptation (TTA) for DAug using EBMs (Xiao et al., 2023; Yuan et al., 2024; Tiwary et al., 2023). E.g., Xiao et al. (2023) first trains EBM on source domain and during testing, they use LD to convert test samples using these trained EBMs. However, this method has a fundamental flaw akin to that of data coverage problem faced in score-based models (Song & Ermon, 2019). Specifically, since the trained EBMs have never seen the test data, the assigned energy to test data manifold could be arbitrary. This renders TTA unreliable for domain adaptation.

Feature disentanglement approaches (Zunino et al., 2021; Peng et al., 2020; Ilse et al., 2020; Qiao et al., 2020; Zhang et al., 2021; Mahajan et al., 2021) aim to separate representations into domain-invariant and domain-specific components, allowing combinations that yield diverse representations for downstream tasks. Mixstyle (Zhou et al., 2021) facilitates generalization by blending instance-level feature statistics across domains, generating novel styles in feature space. RAM (Zhou et al., 2022b) promotes segmentation generalization by mixing amplitude spectra from different domains and incorporating a domain-specific image restoration module. TriD (Chen et al., 2023) supports generalization in 2D medical image segmentation through uniform sampling and channel-wise style mixing of feature statistics, broadening feature space coverage and enhancing robustness. Our work focuses on this promising stream of data manipulation methods and propose a novel approach called Langevin Data Augmentation (LangDAug) for domain generalization.

### 2.2. Data Manipulation based DG

Owing to the difficulty in obtaining large datasets, data manipulation-based domain generalization (DG) methods

(Xu et al., 2021; Liu et al., 2022; Peng et al., 2021; Zhang et al., 2020b; Zhou et al., 2020c; Robey et al., 2021; Liu et al., 2018; Fick et al., 2021; Li et al., 2022a; Zhou et al., 2022b; Liu et al., 2021a) are often sought in medical settings, to enhance training set diversity. These methods are categorized (Wang et al., 2022) into (i) data augmentation (DAug) and (ii) data generation. DAug techniques (Honarvar Nazari & Kovashka, 2020; Prakash et al., 2019; Zhou et al., 2020a; 2023) apply some combination of operations such as flipping, rotation, scaling, cropping, or model-based transformations to reduce overfitting, while data generation methods (Rahman et al., 2019; Somavarapu et al., 2020; Zhang, 2017; Zhou et al., 2020b; Li et al., 2021a) utilize generative models to create diverse data that mirrors the training distribution. MTS (Li et al., 2022a) enhances model-agnostic meta-learning for DG via task augmentation through mixed task sampling and a meta-update objective, effectively mitigating task-level overfitting. RandConv(Xu et al., 2021) uses multi-scale random convolutions as a DAug technique to improve model robustness. FedDG (Liu et al., 2021a) adopts a federated DG strategy that preserves privacy by exchanging amplitude spectra between clients and employs continuous frequency space interpolation and boundary-oriented episodic learning to bolster model generalization to unseen domains.

## 3. Proposed Methodology

### 3.1. Problem Setup and Method Overview

We consider the standard domain generalization setup in which, we have $n$ domains (also called *source* domains) $\{\mathcal{D}_i\}_{i=1}^n$ where each domain admits and underlying probability density $\mathcal{D}_i \sim \mathcal{P}_{\mathcal{D}_i}(x,y)$ on a common support $\mathcal{X} \times \mathcal{Y}$. Further, we are given $n_i$ samples from domain $\mathcal{D}_i$ that are independent and identically distributed, $\{\mathbf{x}_j, \mathbf{y}_j\}_{j=1}^{n_i} \overset{\text{i.i.d}}{\sim} \mathcal{P}_{\mathcal{D}_i}(x,y)$. The goal of an ERM-framework is to learn a parametric mapping $f_\theta(\cdot) : \mathcal{X} \to \mathcal{Y}$ such that it minimizes the empirical risk. Particularly, the parameters $\theta$ are obtained via solving the following optimization problem:

$$\arg\min_\theta \; \mathbb{E}\left[\ell\left(f_\theta(\mathbf{x}), \mathbf{y}\right)\right] \approx \arg\min_\theta \frac{1}{N} \sum_{i=1}^N \ell\left(f_\theta(\mathbf{x}_i), \mathbf{y}_i\right)$$

where $N = \sum_i n_i$. Formally, the True risk is approximated with an unbiased estimate using law of large numbers to get the Empirical risk. This is done in order to obtain a tractable objective function which can be minimized using gradient-based optimization methods. However, since the approximation only takes the provided $n$ domains into account, it is not able to generalize on unseen domain (also called *target* domain) $\mathcal{D}_{n+1} \notin \{\mathcal{D}_i\}_{i=1}^n$. The goal of domain generalization is to aleviate this issue by obtaining

parameter $\theta$ that is able to generalize well on unseen domains.

To address this issue, we adopt a data-augmentation-based approach by introducing new samples in the ERM framework to cover broader support in the $\mathcal{X} \times \mathcal{Y}$ domain, improving the True risk estimate. Specifically, we train Energy-Based Models (EBMs) to traverse between source domain pairs using gradient-based Markov Chain Monte Carlo (MCMC) methods, such as Langevin dynamics (LD). The intermediate iterates generated during this process act as 'bridge' samples between domains. We leverage these intermediate samples as augmentation data, referred to as *Langevin-data Augmentation*, to enhance the model's generalization to unseen target domains.

While LangDAug can be potentially used for normal DG scenarios, we note that there are certain factors that guided our application design. The effectiveness of LangDAug depends on the ability of EBMs to effectively traverse and interpolate between source domain distributions. Such traversal becomes especially natural when source domains share structured similarities or consistent underlying factors of variation. Medical imaging data typically demonstrates structured variations, predominantly reflected through differences in amplitude spectra across domains (Zhou et al., 2022b; Liu et al., 2021a). EBMs have been shown to excel at capturing and modeling variations in amplitude spectra (Tancik et al., 2020; Du et al., 2020a), making them suited for the domain variations characteristic of medical imaging data. Thus, LangDAug's capability aligns well with the domain shift challenges encountered specifically in medical image segmentation.

### 3.2. EBMs for Inter-Domain Transversal

We train Energy-Based Models (EBMs) (Du & Mordatch, 2019; Zhao & Chen, 2021) to traverse between pairs of source domains. Consider two distinct domains $\mathcal{D}_i$ and $\mathcal{D}_j$ where $i \neq j$. To train an EBM, $E_{\theta_{ij}}$ that can transverse from $\mathcal{D}_i$ to $\mathcal{D}_j$, we employ the contrastive divergence loss objective (Hinton, 2002), $\mathcal{L}_{CD}$. The gradient of this objective is:

$$\nabla_{\theta_{ij}} \mathcal{L}_{CD} = \underset{\mathcal{P}_{\mathcal{D}_j}}{\mathbb{E}} \left[ \nabla_{\theta_{ij}} E_{\theta_{ij}}(\mathbf{x}) \right] - \underset{\mathcal{P}_{\theta_{ij}}}{\mathbb{E}} \left[ \nabla_{\theta_{ij}} E_{\theta_{ij}}(\mathbf{x}) \right]$$

$$(1)$$

$$\text{where,} \mathcal{P}_{\theta_{ij}} = \frac{\exp\left(-E_{\theta_{ij}}(\mathbf{x})\right)}{\left(Z_{\theta_{ij}} = \int_{\mathcal{X}} \exp\left(-E_{\theta_{ij}}(\mathbf{x})\right) d\mathbf{x}\right)} \quad (2)$$

We approximate the expectations in Eq. 1 using finite sample averages from their respective distributions. Since sampling directly from $\mathcal{P}_{\theta_{ij}}$ is intractable due to the partition function $Z_{\theta_{ij}}$, we employ MCMC methods, specifically Langevin dynamics (LD).

The LD iteration is initialized with a sample from domain

$\mathcal{D}_i$ and then runs for $K$ steps using $E_{\theta_{ij}}$ as the kernel to approximate sampling from $\mathcal{P}_{\theta_{ij}}$. Mathematically,

$$\mathbf{x}_{t+1} = \mathbf{x}_t - \frac{\alpha^2}{2} \nabla_{\mathbf{x}_t} E_{\theta_{ij}}(\mathbf{x}_t) + \alpha \epsilon, \quad (3)$$

where, $\mathbf{x}_0 \sim \mathcal{P}_{\mathcal{D}_i}(x)$. Thus, we learn $\theta_{ij}$ using Eq. 1, where the second term is approximated using samples from LD initialized with a sample from $\mathcal{D}_i$. The LD iteration can be interpreted as starting from a sample under distribution $\mathcal{P}_{\mathcal{D}_i}$ and gradually transforming it into a sample from $\mathcal{P}_{\mathcal{D}_j}$. In other words, the LD updates traverse from domain $\mathcal{D}_i$ to domain $\mathcal{D}_j$. The intermediate samples generated during the LD process can be viewed as forming a bridge between the two domains. We train such an EBM for all $2\binom{n}{2}$ possible pairs within $n$ source domains.

### 3.3. Langevin Data Augmentation

As explained previously, we learn an EBM for transversal between each pair of domains. Here, we explain how to generate the augmentation samples. Consider the EBM $E_{\theta_{ij}}$ which can transverse from domain $\mathcal{D}_i$ to domain $\mathcal{D}_j$. To generate samples for augmentation, we take a sample $\{\mathbf{x}_j, \mathbf{y}_j\} \in \mathcal{D}_i$ and run LD for $K$ steps, starting from $\mathbf{x}_j$:

$$\mathbf{x}_j^{t+1} = \mathbf{x}_j^t - \frac{\beta^2}{2} \nabla_{\mathbf{x}_j^t} E_{\theta_{ij}}(\mathbf{x}_j^t) + \beta \epsilon, \text{ where, } \mathbf{x}_j^0 = \mathbf{x}_j$$

$$(4)$$

The above process gives rise to $K$-intermediate samples $\{\mathbf{x}_j^t\}_{t=1}^K$. We use the samples $\{\mathbf{x}_j^t, \mathbf{y}_j\}_{j=1, t=1}^{n_i, K}$ (named 'Langevin' data) for augmentation during ERM-based learning[1]. On a closer look, we can see that for a given $k$, $\mathcal{D}_{ij}^k \triangleq \{\mathbf{x}_j^k, \mathbf{y}_j\}_{j=1}^{n_i}$ follow the same distribution as they are obtained via $k$-step LD updates. This can be viewed as samples from a new domain altogether. Hence, we can see that we are able to obtain data from new novel domains by transversing between the original source domains. While the original dataset consisted of $D_{src} \triangleq \bigcup_i \mathcal{D}_i$, the augmented dataset consists of $D_{aug} \triangleq \bigcup_{i \neq j, k} \mathcal{D}_{ij}^k$. This shows that we can generate novel data-points by simply transversing between source domains in a meaningful way. We refer the reader to Supplementary for all the implementation details.

## 4. Theoretical Analysis

**Analysis Overview**: Our theoretical analysis reveals how LangDAug naturally induces regularization and improves generalization. We follow steps similar to Arora et al. (2021); Zhang et al. (2020a) by first demonstrating that

---

[1]Note that we retain original labels while augmentation because, we empirically observe that the Langevin samples preserve the content of $\mathbf{x}_j$ (see Supplementary for visualizations).

LangDAug regularizes the directional derivatives of $f_\theta(\cdot)$ for any parametric model (Thm. 4.1). This insight leads us to examine generalized linear models (GLMs), where we characterize these regularization effects more precisely (Cor. 4.2). We then establish bounds on the Rademacher complexity of the resulting function class (Thm. 4.3) and use these to bound the generalization gap between true and empirical risk (Cor. 4.4).

**Notations**: We denote a general parametric loss function $\ell(\theta, \mathbf{z})$, where $\theta \in \Theta \subseteq \mathbb{R}^d$ and $\mathbf{z} = (\mathbf{x}, \mathbf{y})$ denotes the input output pair. Further, we consider the training dataset $\mathcal{D} = \{\mathbf{x}_i, \mathbf{y}_i\}_{i=1}^k \in \mathcal{X} \times \mathcal{Y}$ such that they follow an underlying distribution, $\mathbf{z}_i = (\mathbf{x}_i, \mathbf{y}_i) \overset{\text{i.i.d}}{\sim} \mathcal{P}_{\mathcal{D}}(\cdot)$. Further, we denote $\tilde{\mathbf{x}}_i(\beta) = \mathbf{x}_i - \frac{\beta^2}{2} \nabla_{\mathbf{x}} \log p(\mathbf{x}_i) + \beta \varepsilon$, $\varepsilon \sim \mathcal{N}(0, I)$ which is the sample obtained by running one-step LD starting from $\mathbf{x}_0 \sim \mathcal{P}_{\mathcal{D}}(x)$ (we drop the dependence on $\beta$ for brevity). We denote $\tilde{\mathbf{z}}_i = (\tilde{\mathbf{x}}_i, \mathbf{y}_i)$ as the data pair used for augmentation. Further, we denote the model parameterized by $\theta$ as $f_\theta : \mathcal{X} \to \mathcal{Y}$, the gradient w.r.t $x$ and $\theta$ is denoted by $\nabla f_\theta(x)$ and $\nabla_\theta f_\theta(x)$ respectively.

Lastly, we denote the true risk as $\mathcal{L}(\theta) = \mathbb{E}[\ell(\theta, \mathbf{z})]$ where the expectation is w.r.t true data distribution. The standard empirical and the augmented empirical risks are denoted by:

$$\mathcal{L}^{\text{std}}(\theta, \mathcal{D}) = \frac{1}{k} \sum_i \ell(\theta, \mathbf{z}_i) \tag{5}$$

$$\mathcal{L}^{\text{aug}}(\theta, \mathcal{D}) = \frac{1}{k} \sum_i \mathop{\mathbb{E}}_{\varepsilon \sim \mathcal{N}(0, I)} [\ell(\theta, \tilde{\mathbf{z}}_i)] \tag{6}$$

We start our alalysis by characterizing the regularization effect of LangDAug:

**Theorem 4.1.** *Consider a real-valued loss function of the form $\ell(\theta, (\mathbf{x}, \mathbf{y})) = h(f_\theta(\mathbf{x})) - \mathbf{y}(f_\theta(\mathbf{x}))$ where $h(\cdot)$ and $f_\theta(\cdot)$ are twice differentiable for all $\theta \in \Theta$. Then given a dataset $\mathcal{D} = \{\mathbf{x}_i, \mathbf{y}_i\}_{i=1}^k$, the augmented empirical risk as denoted in Eq. 6 can be written as:*

$$\mathcal{L}^{aug}(\theta, \mathcal{D}) = \mathcal{L}^{std}(\theta, \mathcal{D}) + \sum_{i=1}^3 \mathcal{R}_i(\theta, \mathcal{D}) \tag{7}$$

*where,*

$$\mathcal{R}_1(\theta, \mathcal{D}) = -\frac{1}{k} \sum_{i=1}^k \beta^2 (h'(f_\theta(\mathbf{x}_i)) - \mathbf{y}_i) \nabla f_\theta(\mathbf{x}_i)^T \nabla \log p(\mathbf{x}_i)$$

$$\mathcal{R}_2(\theta, \mathcal{D}) = \frac{1}{k} \sum_{i=1}^k \beta^2 h''(f_\theta(\mathbf{x}_i)) \text{Tr}(\nabla f_\theta(\mathbf{x}_i) \nabla f_\theta(\mathbf{x}_i)^T)$$

$$\mathcal{R}_3(\theta, \mathcal{D}) = \frac{1}{k} \sum_{i=1}^k \beta^2 (h'(f_\theta(\mathbf{x}_i)) - \mathbf{y}_i) \text{Tr}(\nabla^2 f_\theta(\mathbf{x}_i)).$$

The regularization terms reveal how LangDAug controls model behavior through derivatives of the prediction function. To understand these effects more concretely, we examine GLMs, which model predictive densities using exponential families with linear sufficient statistics (Murphy, 2022):

$$p(y|\mathbf{x}, \theta) = r(\mathbf{x}) \exp\left(y\theta^T \mathbf{x} - A(\theta^T \mathbf{x})\right) \tag{8}$$

where $r(\mathbf{x})$ is the base measure and $A(\cdot)$ is the log partition function. The negative log-likelihood takes the form $A(\theta^T \mathbf{x}) - y\theta^T \mathbf{x}$, allowing us to apply Thm. 4.1 to obtain:

**Corollary 4.2.** *For a GLM, if $A(\cdot)$ is twice differentiable, then the regularization terms obtained via second-order approximation is given by:*

$$\mathcal{R}_{GLM} \triangleq \frac{\beta^2}{2n} \sum_{i=1}^k \left(A''(\theta^T \mathbf{x}_i)\theta^T \theta - A'(\theta^T \mathbf{x}_i)\theta^T s(\mathbf{x}_i)\right)$$

*where, $s(\mathbf{x}_i) = \nabla \log p(\mathbf{x}_i)$ is the Stein's score function.*

Having obtained the above regularization term, we use the techniques in Arora et al. (2021); Zhang et al. (2020a) to bound the Rademacher complexity of the following class of functions which are used in optimization of dual of above regularization term:

$$\mathcal{W}_\gamma \triangleq \left\{ \mathbf{x} \to \theta^T \mathbf{x} : \theta^T \mathop{\mathbb{E}}_x \left[A''(\theta^T \mathbf{x})\theta - A'(\theta^T \mathbf{x})s(\mathbf{x})\right] \leq \gamma \right\}$$

The following provides an upper-bound on Rademacher complexity of such class of functions:

**Theorem 4.3.** *Assume that the distribution of $\mathbf{x}_i$ is $\rho$-retentive, and let $\Sigma_x = \mathbb{E}[\mathbf{x}\mathbf{x}^T]$ have bounded singular values. Further assume that the norm of the parameters and $\mathbb{E}_x[\|\nabla \log p(\mathbf{x})\|^2]$ are bounded. Then the empirical Rademacher complexity of $\mathcal{W}_\gamma$ satisfies:*

$$Rad(\mathcal{W}_\gamma, \mathcal{D}) \leq C \sqrt{\frac{rank(\Sigma_x)}{k}} \tag{9}$$

$$\text{where, } C = \left(\frac{\gamma}{\rho}\right)^{1/2} \vee \left(\frac{\gamma}{\rho\sigma}\right)^{1/4} \tag{10}$$

*here, $\sigma$ denotes the lowest singular values of $\Sigma_x$.*

Now, using this result, we can directly use the results of Bartlett & Mendelson (2002) to bound the generalization gap as follows:

**Corollary 4.4.** *If $A(\cdot)$ is $L_A$-Lipchitz continuous, $\mathcal{X}$, $\mathcal{Y}$, $\Theta$ are all bounded, then there exists constant $L, B > 0$ such that for all $\theta$ satisfying the constraint in $\mathcal{W}_\gamma$, we have:*

$$\mathcal{L} \leq \mathcal{L}^{std} + 2LL_A \left(C \sqrt{\frac{rank(\Sigma_x)}{k}}\right) + B \sqrt{\frac{\log(1/\delta)}{2k}}$$

*with probability atleast $1 - \delta$.*

We note that the appearance of $rank(\Sigma_x)$ rather than the full dimensionality of $\mathbf{x}$ suggests that LangDAug's generalization bound depends on how many "true" degrees of freedom the data has (intrinsic dimension) rather than how many numbers we use to store each data point (ambient dimension). The dependence on intrinsic dimensionality rather than ambient dimension is particularly valuable for high-dimensional problems where the data lies on a lower-dimensional manifold (Gorban & Tyukin, 2018; Cayton et al., 2008).

# 5. Experiments

## 5.1. Datasets and Baselines

**Datasets and Metrics**: We evaluate LangDAug's effectiveness under domain shift using two widely-used medical imaging datasets: Retinal Fundus Segmentation (RFS) (Almazroa et al., 2018; Orlando et al., 2020; Zhang et al., 2010; Sivaswamy et al., 2014) and Prostate MRI Segmentation (Liu et al., 2020). For both datasets, we employ a leave-one-out protocol (Gulrajani & Lopez-Paz, 2021) to assess domain generalization performance.

The RFS dataset challenges models to segment the optical cup (OC) and optical disc (OD) in retinal images. It contains samples from four clinical sites, each representing a distinct domain with its own train/test split. We use the training splits for training and resize all images to $256 \times 256$ pixels during preprocessing. Segmentation accuracy is evaluated using Intersection-over-Union (IoU) and Dice Similarity Score (DSC) metrics.

The Prostate MRI dataset includes 116 T2-weighted MRI scans from six clinical sites (domains). Preprocessing involves cropping each scan to the 3D prostate region, extracting 2D axial slices, and resizing them to $384 \times 384$ pixels. We train on these 2D slices and evaluate using: (1) DSC between predicted and ground truth slices, and (2) Average Surface Distance (ASD) for reconstructed 3D volumes, obtained by concatenating model-predicted slices. Results are reported as the mean of three independent runs.

**Baselines**: We evaluate our method against two categories of domain generalization approaches. First, we consider three recent methods from the DomainBed benchmark (Gulrajani & Lopez-Paz, 2021): Hutchinson (Hemati et al., 2023), which matches domain statistics using Hessian information; Fish (Shi et al., 2022), which employs gradient-based domain alignment; and Fishr (Rame et al., 2022), which utilizes gradient variance matching. Further, we compare against recent approaches specifically tailored for medical image segmentation: RandConv (Xu et al., 2021), which leverages random convolution for style invariance; MixStyle (Zhou et al., 2021), which performs feature-level style mixing; FedDG (Liu et al., 2021a), which

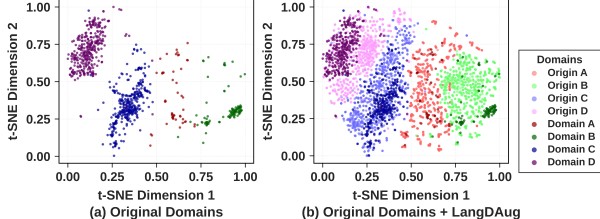

Figure 2: t-SNE visualization comparing domain distributions: (a) the original four domains, and (b) LangDAug generated augmented domains. Origin $x$ denotes samples generated by LangDAug starting from Domain $x$.

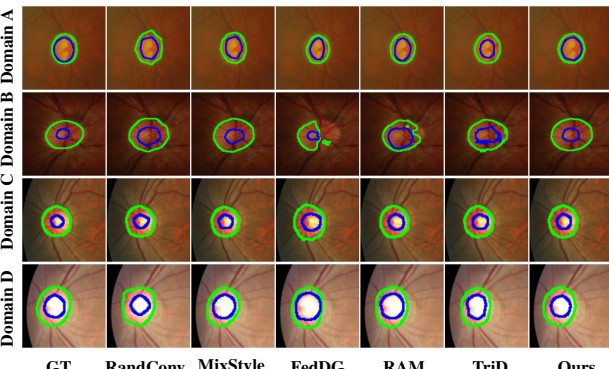

Figure 3: Visualization of retinal fundus segmentation performance of different domain generalization methods.

employs federated learning for domain generalization; and TriD (Chen et al., 2023), which exploits distribution-level information. Since, we follow similar steps as Zhang et al. (2020a) for our analysis[2] we also compare our method with RAM (Zhou et al., 2022b) which is a variant of MixUp designed for medical image segmentation. Additionally, we provide comparisons with CORAL (Sun & Saenko, 2016), RSC (Huang et al., 2020), For a fair comparison we use ResNet34 model (He et al., 2016) as the segmentation backbone for all the methods. All the implementation details including hyperparameters and Ablation studies are provided in Supplementary.

## 5.2. Results

**t-SNE Visualization**: Before presenting the main results, we visualize the RFS dataset samples in the feature space using t-SNE plots (Figure 2). The original domain samples show a concentrated distribution, while Langevin samples significantly expand feature space coverage. Notably, Langevin samples bridge gaps between domains, consistent with the framework in Section 3.2. This inter-domain traversal property allows the model to learn a more comprehensive

---

[2]while the proof steps share similarities, we use different techniques and arrive at an independent verification of the result

representation of the domain space. These observations support our hypothesis on the effectiveness of LangDAug in improving domain generalization. Visualizations of Langevin samples are provided in Supplementary.

### 5.2.1. FUNDUS SEGMENTATION

We provide the results of optical cup (OC) and optical disc (OD) segmentation on retinal fundus images in Table 1. We present the metrics in the form of (OC, OD) for convenience, further, we also provide the mean metric across OC and OD. We also provide visual qualitative comparison of segmentation results in Figure 3. We observe that LangDAug performs most consistently across all domains.

Among the DomainBed approaches (Hutchinson, Fish, and Fishr), performance is generally underwhelming. Hutchinson performed the poorest, with average IoU and DSC of 67.39 and 78.14 respectively. While Fishr showed some improvement over the others, reaching average IoU and DSC of 72.28 and 82.16 respectively. Overall these methods lagged behind other approaches. This pattern was consistent across different domains and metrics.

Domain B emerges as the most challenging for generalization, with most methods failing to achieve mDSC above 80. The reason for this is significantly different conditions under which the fundus images are acquired. In other words, the distribution of Domain B is very different from that of other domains. In this domain, LangDAug demonstrates superior performance with an mIoU of 75.05, surpassing the next best method (RAM) by 1.26%. For optical disc segmentation, LangDAug achieves the highest DSC of 91.64, followed by RAM at 90.52. While RAM leads in optical cup segmentation with a DSC of 82.82, LangDAug maintains competitive performance at 80.11, demonstrating consistent effectiveness across all metrics in this challenging domain. This also shows robustness of LangDAug in limited source distribution setting where source domains are not able to cover the target domain. In such scenarios, previous method fail to generalize effectively whereas LangDAug maintains edge over these methods.

In Domain C, LangDAug demonstrates good performance with the highest mIoU of 81.01, while maintaining competitive mDSC metrics second only to TriD. The method shows balanced effectiveness across both optical cup and disc segmentation, surpassing established approaches like FedDG and RAM. This performance extends to Domain D, where LangDAug leads in both mIoU and mDSC metrics, outperforming comparable methods like TriD and RAM.

While established methods MixStyle, RandConv, and TriD demonstrate marginally better performance in Domain A (mDSC: 88.62, 87.90, and 87.80 respectively), they exhibit substantial performance deterioration in Domain B, with

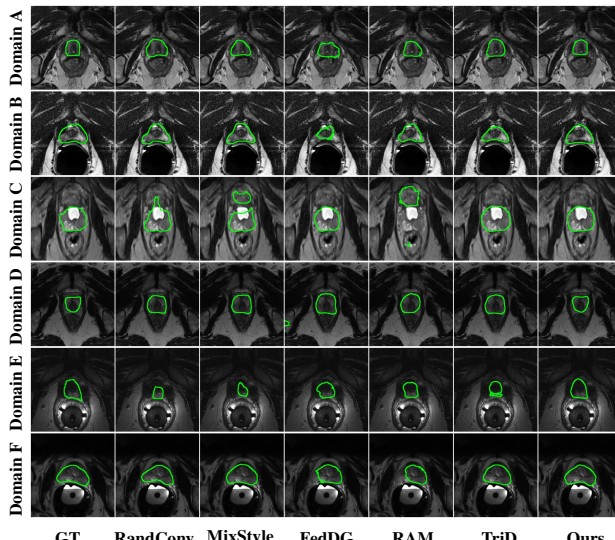

Figure 4: Visualization of prostate segmentation performance of different domain generalization methods.

mDSC values dropping to approximately 78. In contrast, LangDAug maintains remarkable consistency across both domains, achieving competitive mDSC of 86.99 on Domain A while maintaining its performance with an mDSC of 85.87 on Domain B.

LangDAug demonstrates superior overall performance, achieving the highest average IoU (78.84%) and DSC (87.61%) across all domains. Notably, it exhibits better stability in cross-domain performance, as evidenced by significantly lower standard deviations (mIoU: $\pm2.43$, mDSC: $\pm1.89$) compared to its counterparts TriD (mIoU: $\pm3.76$, mDSC: $\pm5.12$) and RAM (mIoU: $\pm2.89$, mDSC: $\pm3.42$). This reduced variance underscores LangDAug's consistent generalization capabilities.

### 5.2.2. PROSTATE SEGMENTATION

Table 2 presents our results for 2D MRI prostate segmentation. We provide visual results in Figure 4. Consistent with the findings from fundus segmentation, the DomainBed methods (Hutchinson, Fish, and Fishr) generally demonstrate lower performance, with Fishr showing the best performance and Hutchinson showing the worst performance among the three.

LangDAug demonstrates best performance across Domains A through D, consistently outperforming the second-best methods in both ASD and DSC. Specifically, in Domains A and B, LangDAug achieves approximately 0.1mm lower ASD and nearly 0.6% higher DSC compared to the next best method, TriD. In Domain D, LangDAug surpasses other methods with a substantial reduction of 0.14mm in ASD and an increase in DSC by over 4%. Even in the

Table 1: Cross-domain generalization performance on Retinal Fundus Segmentation. IoU (%) (↑) and DSC (%) (↑) are presented in pairs $(x, y)$ where, $x$ denotes OC and $y$ denotes OD segmentation metric. Bold ($\boldsymbol{x}$) and Underlined ($\underline{x}$) values indicate best and second-best performance respectively for each metric in their respective columns. Average denotes the average metric across all domains.

| Domain | A | | | | B | | | | C | | | | D | | | | Average | |
|---|---|---|---|---|---|---|---|---|---|---|---|---|---|---|---|---|---|---|
| | IoU | mIoU | DSC | mDSC | IoU | mIoU | DSC | mDSC | IoU | mIoU | DSC | mDSC | IoU | mIoU | DSC | mDSC | IoU | DSC |
| CORAL | (66.59, **90.14**) | 78.37 | (78.79, 94.31) | 86.55 | (54.80, 73.07) | 63.94 | (66.95, 81.46) | 74.20 | (71.34, 86.54) | 78.94 | (82.82, 92.59) | 87.71 | (67.82, 83.87) | 75.85 | (80.19, 91.10) | 85.65 | 74.27 | 83.53 |
| RSC | (64.67, 88.97) | 76.82 | (77.98, 94.24) | 86.11 | (53.34, 71.74) | 62.54 | (64.94, 79.90) | 72.42 | (72.78, 86.67) | 79.73 | (83.53, 92.32) | 87.92 | (68.39, 81.11) | 74.75 | (80.70, 89.38) | 85.04 | 73.46 | 82.87 |
| SagNet | (64.45, 86.38) | 75.42 | (77.34, 92.64) | 84.99 | (46.82, 62.21) | 54.51 | (57.44, 69.94) | 63.69 | (67.29, 83.76) | 75.52 | (79.73, 90.86) | 85.29 | (55.14, 79.48) | 67.31 | (70.04, 88.36) | 79.20 | 68.19 | 78.29 |
| SWAD | (65.32, 87.06) | 76.19 | (77.09, 92.89) | 84.99 | (55.63, 75.01) | 65.32 | (68.21, 83.92) | 76.07 | (67.94, 85.89) | 76.91 | (80.43, 92.17) | 86.30 | (61.57, 82.60) | 72.09 | (75.37, 90.26) | 82.82 | 72.63 | 82.54 |
| Hutchinson | (55.19, 78.27) | 66.73 | (68.10, 86.81) | 77.46 | (55.19, 78.27) | 66.73 | (68.10, 86.81) | 77.46 | (56.17, 82.54) | 69.36 | (70.16, 90.17) | 80.17 | (50.45, 72.36) | 66.73 | (64.42, 83.07) | 77.46 | 67.39 | 78.14 |
| Fish | (63.16, 85.73) | 74.44 | (76.76, 91.16) | 83.96 | (48.91, 74.87) | 64.44 | (64.38, 87.01) | 75.70 | (63.61, 82.92) | 73.26 | (79.99, 88.87) | 84.43 | (56.27, 80.59) | 72.50 | (76.54, 89.94) | 83.24 | 71.16 | 81.83 |
| Fishr | (65.35, 86.35) | 75.85 | (77.33, 91.45) | 84.39 | (54.23, 73.82) | 64.03 | (65.17, 82.32) | 73.75 | (68.32, 82.95) | 75.63 | (80.53, 89.45) | 84.99 | (64.09, 83.11) | 73.60 | (79.99, 91.02) | 85.50 | 72.28 | 82.16 |
| RandConv | (71.44, 88.49) | 79.96 | (81.99, 93.81) | 87.90 | (52.96, 82.28) | 67.62 | (66.35, 90.11) | 78.23 | (71.65, 87.93) | 79.79 | (83.38, 93.39) | 88.38 | (67.56, 86.62) | 77.09 | (80.03, 92.72) | 86.37 | 76.33 | 85.58 |
| MixStyle | (71.76, 89.77) | 80.76 | (82.68, 94.56) | 88.62 | (58.15, 77.23) | 67.69 | (71.17, 86.75) | 78.96 | (71.65, 87.93) | 79.79 | (83.38, 93.39) | 88.38 | (67.56, 86.62) | 77.09 | (80.03, 92.72) | 86.37 | 76.33 | 85.58 |
| FedDG | (**73.30**, 80.01) | 76.65 | (78.52, 84.42) | 81.47 | (68.70, 75.58) | 72.14 | (77.81, 88.35) | 83.08 | (69.92, 82.28) | 76.10 | (83.11, 87.32) | 85.21 | (69.01, 82.91) | 75.96 | (81.63, 88.17) | 84.90 | 75.21 | 83.67 |
| RAM | (67.28, 87.56) | 77.42 | (78.09, 91.66) | 84.87 | (68.92, 78.66) | 73.79 | (**82.82**, 90.52) | 86.67 | (72.21, 86.11) | 79.66 | (81.32, 90.02) | 85.67 | (71.61, 85.88) | 78.74 | (77.17, 91.54) | 84.35 | 77.40 | 85.39 |
| TriD | (71.83, **90.01**) | 80.92 | (79.44, **96.16**) | 87.80 | (68.31, 76.59) | 72.45 | (66.01, 90.42) | 78.21 | (72.66, 86.03) | 79.34 | (**86.22**, 93.83) | 90.02 | (70.52, 87.41) | 78.96 | (83.08, 92.45) | 87.76 | 77.92 | 85.95 |
| LangDAug | (68.31, 89.26) | 78.79 | (79.75, 94.24) | 86.99 | (**70.61**, **79.50**) | 75.05 | (80.11, **91.64**) | 85.87 | (**73.07**, **88.94**) | 81.01 | (83.88, **93.96**) | 88.91 | (**72.33**, **88.70**) | 80.51 | (**83.43**, **93.93**) | 88.68 | 78.84 | 87.61 |

Table 2: Cross-domain generalization performance on T2-weighted Prostate MRI Segmentation. ASD (mm) (↓) and DSC (%) (↑) are presented in pairs $(x, y)$ where, $x$ denotes OC and $y$ denotes OD segmentation metric. Bold ($\boldsymbol{x}$) and Underlined ($\underline{x}$) values indicate best and second-best performance respectively for each metric in their respective columns. Average denotes the average metric across all domains.

| Unseen Domain | A | | B | | C | | D | | E | | F | | Average | |
|---|---|---|---|---|---|---|---|---|---|---|---|---|---|---|
| | ASD | DSC | ASD | DSC | ASD | DSC | ASD | DSC | ASD | DSC | ASD | DSC | ASD | DSC |
| Hutchinson | 3.28 | 82.26 | 1.48 | 85.85 | 2.07 | 72.65 | 3.98 | 77.41 | 2.78 | 71.75 | 1.64 | 81.79 | 2.54 | 78.62 |
| Fish | 1.58 | 84.40 | 1.46 | 85.09 | 3.74 | 70.59 | 3.37 | 78.45 | 3.72 | 74.26 | 0.76 | 83.43 | 2.44 | 79.37 |
| Fishr | 1.10 | 88.03 | 0.94 | 88.41 | 2.07 | 83.21 | 1.08 | 83.94 | 1.78 | 81.41 | 0.61 | 84.41 | 1.26 | 84.90 |
| RandConv | 1.27 | 87.90 | 1.18 | 88.24 | 1.90 | 81.96 | 0.84 | 84.44 | 1.54 | 83.04 | 0.44 | 86.37 | 1.20 | 85.33 |
| MixStyle | 0.72 | 88.59 | 0.88 | 88.42 | 1.62 | 87.10 | 0.65 | 85.62 | 1.59 | 82.18 | 0.51 | 85.72 | 1.00 | 86.27 |
| FedDG | 1.09 | 86.10 | 0.93 | 87.03 | 1.31 | 88.23 | 0.88 | 87.52 | 1.73 | 81.55 | 0.50 | 85.27 | 1.07 | 85.95 |
| RAM | 0.93 | 88.72 | 0.98 | 87.20 | 1.26 | 88.78 | 0.74 | 89.18 | 1.78 | 81.01 | **0.32** | **87.25** | 1.00 | 87.02 |
| TriD | 0.70 | 91.52 | 0.72 | 89.60 | 1.39 | 87.71 | 0.71 | 89.45 | 1.43 | 82.41 | 0.46 | 85.41 | 0.90 | 87.68 |
| LangDAug | **0.58** | **92.15** | **0.64** | **90.31** | 1.21 | **89.53** | **0.57** | **93.41** | 1.49 | **83.17** | 0.38 | 86.41 | **0.81** | **89.16** |

Table 3: Cross-domain generalization performance on Retinal Fundus Segmentation of Domain Randomization method, with and without LangDAug. Values in parentheses for Average columns show absolute improvement over base method.

| Unseen Domain | A | | | | B | | | | C | | | | D | | | | Average | |
|---|---|---|---|---|---|---|---|---|---|---|---|---|---|---|---|---|---|---|
| | IoU | mIoU | DSC | mDSC | IoU | mIoU | DSC | mDSC | IoU | mIoU | DSC | mDSC | IoU | mIoU | DSC | mDSC | mIoU | mDSC |
| FedDG | (73.30, 80.01) | 76.65 | (78.52, 84.42) | 81.47 | (68.70, 75.58) | 72.14 | (77.81, 88.35) | 83.08 | (69.92, 82.28) | 76.10 | (83.11, 87.32) | 85.21 | (69.01, 82.91) | 75.96 | (81.63, 88.17) | 84.90 | 75.21 | 83.67 |
| FedDG + Ours | (74.28, 82.37) | 78.32 | (79.66, 85.29) | 82.47 | (70.03, 76.10) | 73.06 | (79.12, 90.62) | 84.87 | (71.80, 84.41) | 78.10 | (83.03, 88.58) | 86.13 | (70.57, 82.22) | 76.89 | (82.79, 91.37) | 87.08 | 76.59 (+1.38) | 85.14 (+1.47) |
| RAM | (67.28, 87.56) | 77.42 | (78.09, 91.66) | 84.87 | (68.92, 78.66) | 73.79 | (82.82, 90.52) | 86.67 | (72.21, 86.11) | 79.66 | (81.32, 90.02) | 85.67 | (71.61, 85.88) | 78.74 | (77.17, 91.54) | 84.35 | 77.40 | 85.39 |
| RAM + Ours | (70.29, 90.51) | 80.40 | (80.66, 94.76) | 87.71 | (70.38, 80.11) | 75.24 | (83.38, 91.47) | 87.42 | (74.71, 87.99) | 81.35 | (83.84, 92.64) | 88.24 | (73.70, 86.37) | 80.03 | (79.81, 92.89) | 86.35 | 79.26 (+1.86) | 87.43 (+2.04) |
| TriD | (71.83, 90.01) | 80.92 | (79.44, 96.16) | 87.80 | (68.31, 76.59) | 72.45 | (66.01, 90.42) | 78.21 | (72.66, 86.03) | 79.34 | (86.22, 93.83) | 90.02 | (70.52, 87.41) | 78.96 | (83.08, 92.45) | 87.76 | 77.92 | 85.95 |
| TriD + Ours | (72.89, 92.73) | 82.81 | (80.34, 97.82) | 89.08 | (81.67, 78.52) | 80.09 | (73.02, 90.95) | 81.98 | (73.16, 87.93) | 80.54 | (87.32, 96.35) | 91.83 | (71.83, 89.02) | 80.42 | (85.40, 94.09) | 89.74 | 80.97 (+3.05) | 88.16 (+2.21) |

Table 4: Cross-domain generalization performance on 2D MRI Prostate Segmentation of Domain Randomization method, with and without LangDAug. Values in parentheses for Average columns show absolute improvement over base method.

| Unseen Domain | A | | B | | C | | D | | E | | F | | Average | |
|---|---|---|---|---|---|---|---|---|---|---|---|---|---|---|
| | ASD | DSC | ASD | DSC | ASD | DSC | ASD | DSC | ASD | DSC | ASD | DSC | ASD | DSC |
| FedDG | 1.09 | 86.10 | 0.93 | 87.03 | 1.31 | 88.23 | 0.88 | 87.52 | 1.73 | 81.55 | 0.50 | 85.27 | 1.07 | 85.95 |
| FedDG + Ours | 0.90 | 88.24 | 0.66 | 89.31 | 1.16 | 90.22 | 0.75 | 88.19 | 1.68 | 82.03 | 0.48 | 86.10 | 0.94 (−0.13) | 87.35 (+1.40) |
| RAM | 0.93 | 88.72 | 0.98 | 87.20 | 1.26 | 88.78 | 0.74 | 89.18 | 1.78 | 81.01 | 0.32 | 87.25 | 1.00 | 87.02 |
| RAM + Ours | 0.89 | 89.62 | 0.69 | 89.11 | 1.15 | 89.03 | 0.62 | 91.63 | 1.41 | 83.92 | 0.24 | 88.85 | 0.83 (−0.17) | 88.69 (+1.67) |
| TriD | 0.70 | 91.52 | 0.72 | 89.60 | 1.39 | 87.71 | 0.71 | 89.45 | 1.43 | 82.41 | 0.46 | 85.41 | 0.90 | 87.68 |
| TriD + Ours | 0.64 | 92.73 | 0.61 | 91.05 | 1.09 | 92.31 | 0.55 | 94.22 | 1.29 | 85.17 | 0.35 | 87.99 | 0.76 (−0.14) | 90.58 (+2.90) |

challenging Domain C, LangDAug maintains a significant performance advantage, outperforming other methods by 0.18mm in ASD and 1.82% in DSC. Furthermore, in the most difficult Domain E (identified by the highest average ASD and lowest average DSC across methods), LangDAug secures the best DSC, demonstrating its performance consistency across domains.

While RAM achieves the best performance in Domain F (ASD: 0.32 mm, DSC: 87.25%), LangDAug remains competitive in this domain (ASD: 0.38 mm, DSC: 86.41%) and demonstrates the most consistent performance across all domains. This consistency is highlighted by LangDAug's

best overall average performance across domains, which is lower by 0.09 mm in ASD and higher by 1.48% in DSC compared to the second-best TriD, similar to its trend in Fundus segmentation results.

### 5.3. LangDAug with Domain Randomization

LangDAug, described in Section 3, generates samples from intermediate domains using Langevin dynamics. This can enhance domain randomization methods, which combine domain-invariant (DI) and domain-specific (DS) features from source domains. The key assumption is that test domain features lie within the feature space formed by mixing DI-DS features from source domains. Thus, more source domains produce more diverse features, improving target domain performance. By creating new domains via Langevin dynamics, LangDAug enhances domain randomization methods. We validate this on retinal fundus and prostate segmentation datasets, testing with three leading methods: FedDG (Liu et al., 2021a), RAM (Zhou et al., 2022b), and TriD (Chen et al., 2023).

**Fundus Segmentation**: Table 3 presents the performance comparison of domain randomization methods augmented with LangDAug on fundus segmentation. LangDAug demonstrates consistent performance improvements across all baseline methods and domains. Specifically, FedDG augmented with LangDAug shows significant enhancements of +1.38% in average IoU and +1.47% in average DSC across all domains. RAM exhibits even more substantial gains, with improvements of +1.86% and +2.04% in average IoU and DSC respectively, notably achieving an mDSC increase from 84.87% to 87.71% on Domain A. Most remarkably, TriD combined with LangDAug demonstrates the highest improvements of +3.05% and +2.21% in average IoU and DSC respectively, with particularly significant enhancement in Domain B (mDSC increase from 78.21% to 81.98%).These consistent improvements across methods and domains demonstrate LangDAug 's effectiveness in enhancing domain randomization methods.

**Prostate Segmentation**: Table 4 demonstrates LangDAug's effectiveness in enhancing cross-domain generalization for prostate segmentation. Across all six domains, LangDAug consistently improves both ASD and DSC metrics, paralleling the improvements observed in fundus segmentation. The most substantial improvements are observed with TriD, yielding average ASD reduction of 0.14mm and DSC improvement of +2.90%. RAM augmented with LangDAug shows similar gains, with average ASD and DSC improvements of −0.17mm and +1.67% respectively, particularly notable in Domain E where ASD decreased from 1.78 to 1.41mm. FedDG, despite moderate baseline performance, demonstrates significant improvement when combined with LangDAug , especially in Domain B (ASD reduction: 0.93

to 0.66mm, DSC increase: 87.03% to 89.31%), with overall metrics improving by −0.13mm ASD and +1.40% DSC. Notably, while Domain E looks most challenging across all methods, LangDAug consistently enhances performance, underscoring its efficacy in challenging cross-domain scenarios.

These observations conclusively show that LangDAug significantly boosts the performance of all domain randomization baselines.

## 6. Conclusion

In this work, we propose LangDAug, a novel Langevin data augmentation method for multi-source domain generalization in 2D medical image segmentation. LangDAug leverages Energy-Based Models (EBMs) trained via contrastive divergence to generate intermediate samples between source domains using Langevin dynamics. These samples act as bridges across domain distributions. We establish a theoretical foundation for LangDAug, showing its induced regularization effect on parametric models. For Generalized Linear Models (GLMs), we prove that LangDAug's regularization terms upper-bound the Rademacher complexity based on the data manifold's intrinsic dimension. Comprehensive experiments on retinal fundus and prostate MRI segmentation demonstrate LangDAug's superiority over existing methods. Moreover, LangDAug effectively complements domain randomization approaches, achieving state-of-the-art performance in domain generalization tasks.

**Limitations**: While LangDAug delivers strong performance, it has two primary limitations. First, the larger number of training samples results in longer training times, though selective sampling can mitigate this. Second, the number of required EBMs scales with the source domains, posing scalability challenges. Future work could address this by adopting shared architectures with domain conditioning. We provide detailes quantitative computational cost in Supplementary. Further, we process 3D volumes as 2D slices. Future work could explore training EBMs directly on 3D data to bypass slicing and better model 3D spatial relationships.

## Acknowledgements

This work was supported (in part for setting up the GPU compute) by the Indian Institute of Science through a start-up grant. Piyush is supported by Government of India via Prime Minister's Research Fellowship. Kinjawl is supported by Pre-doctoral fellowship provided by Kotak IISc AI-ML Center (KIAC). Prathosh is supported by Infosys Foundation Young investigator award.

## Impact Statement

LangDAug advances domain generalization in medical image segmentation by enabling robust adaptation to diverse imaging environments, a critical challenge in deploying AI models across clinical settings. By generating domain-bridging samples through Langevin dynamics and Energy-Based Models, the method enhances segmentation accuracy in scenarios affected by domain shifts (e.g., variations in scanners or protocols), directly supporting more reliable diagnostics and treatment planning. Its theoretical grounding in bounding model complexity via intrinsic data dimensionality offers a principled framework for future research.

While computational costs and scalability limitations currently constrain its adoption, LangDAug's compatibility with domain randomization and its ability to preserve anatomical fidelity position it as a versatile tool for real-world applications. Future efforts to optimize training efficiency and extend the framework to 3D volumetric data could further unlock its potential, paving the way for AI systems that generalize seamlessly across evolving medical imaging technologies.

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

# A. Proof of Theoretical Results

**Theorem A.1.** *Consider a loss function of the form $\ell(\theta, (\mathbf{x}, \mathbf{y})) = h(f_\theta(\mathbf{x})) - \mathbf{y}(f_\theta(x))$ where $h(\cdot)$ and $f_\theta(\cdot)$ are twice differentiable for all $\theta \in \Theta$. Then given a dataset $\mathcal{D} = \{\mathbf{x}_i, \mathbf{y}_i\}_{i=1}^k$, the augmented empirical risk as denoted in Eq. 6 can be written as:*

$$\mathcal{L}^{aug}(\theta, \mathcal{D}) = \mathcal{L}^{std}(\theta, \mathcal{D}) + \sum_{i=1}^{3} \mathcal{R}_i(\theta, \mathcal{D}) \qquad (11)$$

*where,*

$$\mathcal{R}_1(\theta, \mathcal{D}) = -\frac{1}{k}\sum_{i=1}^{k} \beta^2 (h'(f_\theta(\mathbf{x}_i)) - \mathbf{y}_i)\nabla f_\theta(\mathbf{x}_i)^T \nabla \log p(\mathbf{x}_i)$$

$$\mathcal{R}_2(\theta, \mathcal{D}) = \frac{1}{k}\sum_{i=1}^{k} \beta^2 h''(f_\theta(\mathbf{x}_i))\operatorname{Tr}(\nabla f_\theta(\mathbf{x}_i)\nabla f_\theta(\mathbf{x}_i)^T)$$

$$\mathcal{R}_3(\theta, \mathcal{D}) = \frac{1}{k}\sum_{i=1}^{k} \beta^2 (h'(f_\theta(\mathbf{x}_i)) - \mathbf{y}_i)\operatorname{Tr}(\nabla^2 f_\theta(\mathbf{x}_i))$$

*Proof.* By definition, we have:

$$\mathcal{L}^{aug}(\theta, \mathcal{D}) = \frac{1}{k}\sum_i \mathop{\mathbb{E}}_{\varepsilon \sim \mathcal{N}(0, I)}[\ell(\theta, \tilde{\mathbf{z}}_i)] \qquad (12)$$

Consider one term of the above summation:

$$\ell(\theta, \tilde{\mathbf{z}}_i) = h(f_\theta(\tilde{\mathbf{x}}_i(\beta))) - \mathbf{y}_i(f_\theta(\tilde{\mathbf{x}}_i(\beta))) \triangleq \psi_i(\beta) \quad (13)$$

$$= \psi_i(0) + \beta\psi_i'(0) + \frac{\beta^2}{2}\psi_i''(0) + \beta^2\varphi(\beta) \quad (14)$$

where, we have used Taylor expansion to approximate $\psi_i(\cdot)$ around $\beta = 0$ and $\lim_{\beta \to 0} \varphi(\beta) = 0$. We will evaluate each of the above terms below.
Recall that $\tilde{\mathbf{x}}_i(\beta) = \mathbf{x}_i - \frac{\beta^2}{2}\nabla_{\mathbf{x}} \log p(\mathbf{x}_i) + \beta\varepsilon$, hence, $\tilde{\mathbf{x}}_i(0) = \mathbf{x}_i$. Using this, we have:

$$\psi_i(0) = h(f_\theta(\mathbf{x}_i)) - \mathbf{y}_i(f_\theta(\mathbf{x}_i)) = \ell(\theta, \mathbf{z}_i) \qquad (15)$$

Now,

$$\psi_i'(\beta) = h'(f_\theta(\tilde{\mathbf{x}}_i))\left(\frac{\partial f_\theta(\tilde{\mathbf{x}}_i)}{\partial \tilde{\mathbf{x}}_i}\right)^T \frac{\partial \tilde{\mathbf{x}}_i}{\partial \beta}$$
$$- \mathbf{y}_i\left(\frac{\partial f_\theta(\tilde{\mathbf{x}}_i)}{\partial \tilde{\mathbf{x}}_i}\right)^T \frac{\partial \tilde{\mathbf{x}}_i}{\partial \beta} \qquad (16)$$

$$= \left[h'(f_\theta(\tilde{\mathbf{x}}_i)) - \mathbf{y}_i\right]\nabla f_\theta(\tilde{\mathbf{x}}_i)^T\left[\varepsilon - \beta\nabla \log p(\mathbf{x}_i)\right] \quad (17)$$

$$= \left[h'(f_\theta(\tilde{\mathbf{x}}_i)) - \mathbf{y}_i\right]\nabla f_\theta(\tilde{\mathbf{x}}_i)^T r_\varepsilon(\beta) \qquad (18)$$

where, $r_\varepsilon(\beta) \triangleq \varepsilon - \beta\nabla \log p(\mathbf{x}_i) = \partial\tilde{\mathbf{x}}_i/\partial\beta$. Hence, we have:

$$\psi_i'(0) = (h'(f_\theta(\mathbf{x}_i)) - \mathbf{y}_i)\nabla f_\theta(\mathbf{x}_i)^T \varepsilon \qquad (19)$$

$$\implies \mathop{\mathbb{E}}_{\varepsilon}[\psi_i'(0)] = (h'(f_\theta(\mathbf{x}_i)) - \mathbf{y}_i)\nabla f_\theta(\mathbf{x}_i)^T \mathop{\mathbb{E}}_{\varepsilon}[\varepsilon] = 0$$
$$(20)$$

Next, for the second derivative, we differentiate the above expression:

$$\psi_i''(\beta) = h''(f_\theta(\tilde{\mathbf{x}}_i))\left(\nabla f_\theta(\tilde{\mathbf{x}}_i)^T r_\varepsilon(\beta)\right)^2$$
$$+ \left[h'(f_\theta(\tilde{\mathbf{x}}_i)) - \mathbf{y}_i\right]\left[\varepsilon^T \nabla^2 f_\theta(\tilde{\mathbf{x}}_i)r_\varepsilon(\beta)\right]$$
$$- \left[h'(f_\theta(\tilde{\mathbf{x}}_i)) - \mathbf{y}_i\right]\left[\beta\nabla \log p(\mathbf{x}_i)^T \nabla^2 f_\theta(\tilde{\mathbf{x}}_i)r_\varepsilon(\beta)\right.$$
$$\left. + \nabla \log p(\mathbf{x}_i)^T \nabla f_\theta(\tilde{\mathbf{x}}_i)\right] \qquad (21)$$

$$= h''(f_\theta(\tilde{\mathbf{x}}_i))r_\varepsilon(\beta)^T \left(\nabla f_\theta(\tilde{\mathbf{x}}_i)\nabla f_\theta(\tilde{\mathbf{x}}_i)^T\right)r_\varepsilon(\beta)$$
$$+ \left[h'(f_\theta(\tilde{\mathbf{x}}_i)) - \mathbf{y}_i\right]r_\varepsilon(\beta)^T \left(\nabla^2 f_\theta(\tilde{\mathbf{x}}_i)\right)r_\varepsilon(\beta)$$
$$- \left[h'(f_\theta(\tilde{\mathbf{x}}_i)) - \mathbf{y}_i\right]\nabla f_\theta(\tilde{\mathbf{x}}_i)^T \nabla \log p(\mathbf{x}_i)$$
$$(22)$$

Substituting $\beta = 0$ in the above gives:

$$\psi_i''(0) = h''(f_\theta(\mathbf{x}_i))\varepsilon^T \left(\nabla f_\theta(\mathbf{x}_i)\nabla f_\theta(\mathbf{x}_i)^T\right)\varepsilon$$
$$+ \left[h'(f_\theta(\mathbf{x}_i)) - \mathbf{y}_i\right]\varepsilon^T \left(\nabla^2 f_\theta(\mathbf{x}_i)\right)\varepsilon$$
$$- \left[h'(f_\theta(\mathbf{x}_i)) - \mathbf{y}_i\right]\nabla f_\theta(\mathbf{x}_i)^T \nabla \log p(\mathbf{x}_i) \quad (23)$$

Taking expectation w.r.t $\varepsilon$:

$$\mathop{\mathbb{E}}_{\varepsilon}[\psi_i''(0)] = h''(f_\theta(\mathbf{x}_i))\operatorname{Tr}(\nabla f_\theta(\mathbf{x}_i)\nabla f_\theta(\mathbf{x}_i)^T)$$
$$+ (h'(f_\theta(\mathbf{x}_i)) - \mathbf{y}_i)\operatorname{Tr}(\nabla^2 f_\theta(\mathbf{x}_i))$$
$$- (h'(f_\theta(\mathbf{x}_i)) - \mathbf{y}_i)\nabla f_\theta(\mathbf{x}_i)^T \nabla \log p(\mathbf{x}_i)$$
$$(24)$$

Now, combining results from Eq. 15, 20, 24 into Eq. 14 and Eq. 6:

$$\mathcal{L}^{aug}(\theta, \mathcal{D}) = \frac{1}{k}\sum_i \Big\{ \ell(\theta, \mathbf{z}_i)$$
$$- \frac{\beta^2}{2}(h'(f_\theta(\mathbf{x}_i)) - \mathbf{y}_i)\nabla f_\theta(\mathbf{x}_i)^T \nabla \log p(\mathbf{x}_i)$$
$$+ \frac{\beta^2}{2}h''(f_\theta(\mathbf{x}_i))\operatorname{Tr}(\nabla f_\theta(\mathbf{x}_i)\nabla f_\theta(\mathbf{x}_i)^T)$$
$$+ \frac{\beta^2}{2}(h'(f_\theta(\mathbf{x}_i)) - \mathbf{y}_i)\operatorname{Tr}(\nabla^2 f_\theta(\mathbf{x}_i))\Big\} \quad (25)$$

$$\implies \mathcal{L}^{aug}(\theta, \mathcal{D}) = \mathcal{L}^{std}(\theta, \mathcal{D}) + \sum_{i=1}^{3} \mathcal{R}_i(\theta, \mathcal{D})$$
$$(26)$$

$\square$

**Corollary A.2.** *For a GLM, if $A(\cdot)$ is twice differentiable, then the regularization terms obtained via second-order approximation is given by:*

$$\frac{\beta^2}{2n}\sum_{i=1}^{k} \left(A''(\theta^T \mathbf{x}_i)\theta^T\theta - A'(\theta^T \mathbf{x}_i)\theta^T s(\mathbf{x}_i)\right) \quad (27)$$

*where, $s(\mathbf{x}_i) = \nabla \log p(\mathbf{x}_i)$ is the Stein's score function.*

*Proof.* Using the result of Theorem 4.1 with $h(\cdot) = A(\cdot)$ and $f_\theta(\mathbf{x}) = \theta^T \mathbf{x}$, we get:

$$\ell(\theta, \tilde{\mathbf{z}}_i) = \ell(\theta, \mathbf{z}_i) - \frac{\beta^2}{2} A'(\theta^T \mathbf{x}_i) \theta^T \nabla \log p(\mathbf{x}_i)$$
$$+ \frac{\beta^2}{2} A''(\theta^T \mathbf{x}_i) \theta^T \theta \qquad (28)$$

where we use the fact that for GLMs, $\nabla^2 f_\theta(\cdot) = 0$ and $\mathrm{Tr}(\theta \theta^T) = \theta^T \theta$. This completes the proof. $\qquad \square$

**Theorem A.3.** *Assume that the distribution of $\mathbf{x}_i$ is $\rho$-retentive, and let $\Sigma_x = \mathbb{E}[\mathbf{x}\mathbf{x}^T]$ have bounded singular values. Further assume that the norm of the parameters and $\mathbb{E}_x[\|\nabla \log p(\mathbf{x})\|^2]$ are bounded. Then the empirical Rademacher complexity of $\mathcal{W}_\gamma$ satisfies:*

$$Rad(\mathcal{W}_\gamma, \mathcal{D}) \leq C \sqrt{\frac{rank(\Sigma_x)}{k}} \qquad (29)$$

$$where, \quad C = \left(\frac{\gamma}{\rho}\right)^{1/2} \vee \left(\frac{\gamma}{\rho\sigma}\right)^{1/4} \qquad (30)$$

*here, $\sigma$ denotes the lowest singular values of $\Sigma_x$.*

*Proof.* The function class under consideration is:

$$\mathcal{W}_\gamma \triangleq \left\{ \mathbf{x} \to \theta^T \mathbf{x} : \theta^T \mathbb{E}_x \left[ A''(\theta^T \mathbf{x})\theta - A'(\theta^T \mathbf{x})s(\mathbf{x}) \right] \leq \gamma \right\}$$

The above constraint can be written as:

$$\gamma \geq \mathbb{E}_x[A''(\theta^T \mathbf{x})]\theta^T \theta - \mathbb{E}_x[A'(\theta^T \mathbf{x})\theta^T s(\mathbf{x})] \qquad (31)$$
$$\geq \mathbb{E}_x[A''(\theta^T \mathbf{x})]\theta^T \theta - \sqrt{\mathbb{E}_x(A'(\theta^T \mathbf{x}))^2}\sqrt{\mathbb{E}_x(\theta^T s(\mathbf{x}))^2} \qquad (32)$$
$$\geq \mathbb{E}_x[A''(\theta^T \mathbf{x})]\theta^T \theta - \sqrt{\mathbb{E}_x(A'(\theta^T \mathbf{x}))^2}\sqrt{\|\theta\|^2 \mathbb{E}_x[\|s(\mathbf{x})\|^2]} \qquad (33)$$
$$\geq \mathbb{E}_x[A''(\theta^T \mathbf{x})]\theta^T \theta - \kappa_1 \|\theta\| \sqrt{\mathbb{E}_x(A'(\theta^T \mathbf{x}))^2} \qquad (34)$$
$$\geq \mathbb{E}_x[A''(\theta^T \mathbf{x})]\theta^T \theta - \frac{\kappa_1}{\kappa_2} \|\theta\|^2 \sqrt{\mathbb{E}_x(A'(\theta^T \mathbf{x}))^2} \qquad (35)$$
$$\geq \|\theta\|^2 \left( \mathbb{E}_x[A''(\theta^T \mathbf{x})] - \frac{\kappa_1}{\kappa_2} \sqrt{\mathbb{E}_x(A'(\theta^T \mathbf{x}))^2} \right) \qquad (36)$$

where, $\mathbb{E}_x[\|s(\mathbf{x})\|^2] \leq \kappa_1$ and $\|\theta\|^2 \geq \kappa_2$. From $\rho$-retentiveness, we have:

$$\rho \min\{1, \mathbb{E}_x(\theta^T \mathbf{x})^2\} \leq \mathbb{E}_x \left[ A''(\theta^T \mathbf{x}) - \frac{\kappa_1}{\kappa_2} A'(\theta^T \mathbf{x}) \right] \qquad (37)$$

$$\rho \min\{1, \theta^T \Sigma_x \theta\} \leq \mathbb{E}_x \left[ A''(\theta^T \mathbf{x}) \right] - \frac{\kappa_1}{\kappa_2} \sqrt{\mathbb{E}_x \left[ A'(\theta^T \mathbf{x})^2 \right]} \qquad (38)$$

Combining this with constraint in Eq. 36, we get:

$$\|\theta\|^2 \leq \frac{\gamma}{\rho \min\{1, \theta^T \Sigma_x \theta\}} \leq \max\left\{ \frac{\gamma}{\rho}, \frac{\gamma}{\rho\theta^T \Sigma_x \theta} \right\} \qquad (39)$$

$$\implies \|\theta\|^2 \leq \frac{\gamma}{\rho} \vee \sqrt{\frac{\gamma}{\rho\sigma}} \qquad (40)$$

where $\sigma$ is the lowest singular value of $\Sigma_x$. Now, the empirical Rademacher complexity is given by:

$$Rad(\mathcal{W}_\gamma, \mathcal{D}) = \mathbb{E}_\xi \left[ \sup_{\mathcal{W}_\gamma} \frac{1}{k} \sum_i \xi_i \theta^T \mathbf{x}_i \right] \qquad (41)$$

$$\leq \mathbb{E}_\xi \left[ \sup_{\|\theta\|^2 \leq \frac{\gamma}{\rho} \vee \sqrt{\frac{\gamma}{\rho\sigma}}} \frac{1}{k} \sum_i \xi_i \theta^T \mathbf{x}_i \right] \qquad (42)$$

$$\leq \frac{1}{k} \left(\frac{\gamma}{\rho}\right)^{1/2} \vee \left(\frac{\gamma}{\rho\sigma}\right)^{1/4} \mathbb{E}_\xi \left[ \| \sum_{i=1} \xi_i \mathbf{x}_i \| \right] \qquad (43)$$

$$\leq \frac{1}{k} C \sqrt{\mathbb{E}_\xi \| \sum_i \xi_i \mathbf{x}_i \|^2} \qquad (44)$$

$$\leq \frac{1}{k} C \sqrt{\sum_i \mathbf{x}_i^T \mathbf{x}_i} \qquad (45)$$

Taking expectation over dataset distribution completes the proof:

$$Rad(\mathcal{W}_\gamma, \mathcal{D}) = \mathbb{E}_{\mathcal{D}}[Rad(\mathcal{W}_\gamma, \mathcal{D})] \leq \frac{C}{k} \sqrt{\sum_i \mathbb{E}[\mathbf{x}_i^T \mathbf{x}_i]} \qquad (46)$$

$$\leq \frac{C}{k} \sqrt{\sum_i \mathbb{E}[\mathrm{Tr}(\mathbf{x}_i \mathbf{x}_i^T)]} \qquad (47)$$

$$\leq \frac{C}{k} \sqrt{\sum_i \mathrm{Tr}(\Sigma_x)} \qquad (48)$$

$$\leq C \sqrt{\frac{rank(\Sigma_x)}{k}} \qquad (49)$$

$$\square$$

**Lemma A.4** (Bartlett & Mendelson (2002)). *For any $B$-uniformly bounded and $L$-Lipchitz function $\zeta$, for all $\phi \in \Phi$, with probability atleast $1 - \delta$:*

$$\mathbb{E}[\zeta(\phi(\mathbf{x}))] \leq \frac{1}{k} \sum_i \zeta(\phi(\mathbf{x}_i)) + 2L Rad(\Phi, \mathcal{D}) + B\sqrt{\frac{\log(1/\delta)}{2k}} \qquad (50)$$

**Corollary A.5.** *If $A(\cdot)$ is $L_A$-Lipchitz continuous, $\mathcal{X}, \mathcal{Y}, \Theta$ are all bounded, then there exists constant $L, B > 0$ such that for all $\theta$ satisfying the constraint in $\mathcal{W}_\gamma$, we have:*

$$\mathcal{L} \le \mathcal{L}^{std} + 2LL_A \left( C\sqrt{\frac{rank(\Sigma_x)}{k}} \right) + B\sqrt{\frac{\log(1/\delta)}{2k}} \tag{51}$$

*with probability atleast $1 - \delta$.*

*Proof.* The results follow directly from Lemma A.4 and result of Theorem 4.3. □

## B. Implementation Details

This section outlines the implementation specifics of LangDAug. The code and relevant resources are available at https://github.com/backpropagator/LangDAug.

**Energy-based Models**: LangDAug leverages trained Energy-Based Models (EBMs) to generate augmented samples, as described in the main text. Following the approach in (Zhao & Chen, 2021), we train EBMs in the latent space of a VQ-VAE 2 model[3]. The energy model is implemented using the discriminator architecture of StyleGAN2[4].

First, the VQ-VAE 2 model is trained on all source domains. Subsequently, EBMs are trained in the latent space of this VQ-VAE 2 model to enable domain traversal using Langevin dynamics. The VQ-VAE 2 configuration includes a codebook dimension of 32 and a codebook size of 256. Similar to (Zhao & Chen, 2021), we train the VQ-VAE using only the reconstruction loss, omitting the second-stage PixelCNN training. The EBMs are optimized using the Adam optimizer with a learning rate of 0.001. During domain traversal, the Langevin step size ($\beta$ in Eq. 4) is set to $\beta = 1$, and the number of Langevin steps ($K$) is set to $K = 40$.

**LangDAug**: For $n$ source domains, we train $2\binom{n}{2}$ EBMs to model domain pairs in both directions[5]. To generate Langevin samples, as outlined in Section 3.3, we execute Langevin Dynamics (LD) and store intermediate samples as per Eq. 4. Instead of retaining all $K = 40$ intermediate samples, we store a subset of these samples for two reasons: (a) to manage storage and computational overhead, and (b) to reduce the correlation between MCMC

samples. Specifically, we store 13 samples for fundus segmentation (uniformly at iterations $k = 3, 6, \ldots, 39$)[6] and 5 samples for 2D MRI prostate segmentation (at iterations $k = 5, 10, \ldots, 40$). These samples are precomputed for each EBM to save computational time in downstream segmentation network training. We provide a pseudo-code in Algorithm 1.

**Segmentation Network**: For downstream segmentation, we employ a ResNet34-based network (He et al., 2016)[7]. It comprises of a ResNet34 encoder and ASPP-based decoder. The network is trained using a combination of cross-entropy and DICE loss, with the AdamW optimizer. The learning rate (lr) is selected from $\{1 \times 10^{-4}, 1 \times 10^{-6}\}$, and the batch size (bs) is chosen from $\{8, 32, 64\}$. The running average parameters are set to $\beta_1 = 0.9$ and $\beta_2 = 0.99$. Empirically, we observed that (lr $= 1 \times 10^{-4}$, bs $= 8$) yields optimal performance for fundus segmentation, whereas (lr $= 1 \times 10^{-6}$, bs $= 32/64$) works best for 2D MRI prostate segmentation.

**Baselines**: For all the baselines, we use the same ResNet34-based segmentation network. The DomainBed methods (Fish, Fishr, and Hutchinson) are primarily used in classification tasks. For segmentation, we re-implemented these methods to ensure compatibility with the retinal fundus and prostate datasets, utilizing the ResNet-34 backbone. These adaptations were closely aligned with the original implementations provided in DomainBed[8]. In addition to the default implementation, we re-scaled the losses for better stability and training of network.

We use the publicly available codebase for RandConv[9], MixStyle[10], FedDG[11], RAM[12] and TriD[13]. For MixStyle, we add additional layers after the first two ResNet blocks of segmentation model to transfer the instance level feature statistics between source domains. Further, for RAM and FedDG, instead of UNet encoder-decoder, we use the ResNet34 blocks and ASPP decoder for training.

---

[3]Implementation taken from https://github.com/rosinality/vq-vae-2-pytorch

[4]Implementation taken from https://github.com/rosinality/stylegan2-pytorch

[5]As mentioned in limitations, this can be problematic if the number of source domains is too large. In that case, shared architectures can be explored with conditioning on desired domains

[6]For RFS dataset, we use an $L$-channel replacement strategy to maintain the position of optical disc and optical cup while running LD. Specifically, after each LD step, we replace the $L$ channel of the updated sample with $L$ channel of original sample. This ensures that the position of OC and OD are preserved and segmentation masks remain valid.

[7]Implementation taken from https://github.com/kazuto1011/deeplab-pytorch/tree/master/libs/models

[8]https://github.com/facebookresearch/DomainBed/tree/main

[9]https://github.com/wildphoton/RandConv

[10]https://github.com/KaiyangZhou/mixstyle-release

[11]https://github.com/liuquande/FedDG-ELCFS

[12]https://github.com/zzzqzhou/RAM-DSIR/tree/main

[13]https://github.com/Chen-Ziyang/TriD

**Datasets**: The datasets can be found publicly. The Retinal Fundus Dataset can be found at: https://drive.google.com/file/d/1p33nsWQaiZMAgsruDoJLyatoq5XAH-TH/view, 2D MRI Prostate Dataset can be found at: https://liuquande.github.io/SAML/. We have used the same convention as earlier works (Liu et al., 2021a; Zhou et al., 2022b; Chen et al., 2023) for naming the datasets.

## C. Additional Results

### C.1. Ablation Studies on EBM Hyperparameters

We analyze the impact of various Energy-Based Model (EBM) hyperparameters on cross-domain generalization for the Retinal Fundus Segmentation task. The results are summarized below:

- **Langevin Steps ($K$):** First row of Figure 5 shows the performance across different numbers of Langevin steps used during EBM sampling. LangDAug maintains stable performance at higher values of $K$.

- **Step Size ($\beta$):** Second row of Figure 5 illustrates the effect of varying the step size in Langevin dynamics. Lower values of $\beta$ lead to more stable and reliable performance.

- **EBM Complexity (`#conv_blocks`):** We vary the number of convolutional blocks in the EBM to control model complexity. As shown in the third row of Figure 5, performance remains consistent, suggesting that even lightweight EBMs are sufficient.

- **Number of Augmented Samples (`#samples/chain`):** Fourth row of Figure 5 examines the number of intermediate samples stored per Langevin chain. Too few samples fail to capture vicinal distributions, while too many introduce high auto-correlation between langevin samples leading to biased learning. A moderate, well-spaced selection of samples yields the best results.

### C.2. Computational Cost Analysis

We acknowledge the increased computational cost of LangDAug in our limitations. In Table 5, we provide a detailed comparison of training time and peak memory usage across methods on the Retinal Fundus Segmentation (RFS) task. All experiments were conducted on a single NVIDIA A6000 GPU with 48GB memory, and results are averaged across domains.

Although LangDAug increases training time, it remains comparable to methods like RAM (2.75 hrs) and is notably

---

**Algorithm 1 Langevin Data Augmentation**

**Input**: Source domains $\{\mathcal{D}_i\}_{i=1}^n$, Langevin step size $\beta$, Number of Langevin steps $K$.

**Output**: Augmentation datasets $\{\mathcal{D}_{ij}^k\}_{i=1,j=1,k=1}^{n,n,K}$.

1: Initialize parameters $\{\theta_{ij}\}_{i,j=1}^n$, where $i \neq j$.

---

2: Energy-Based Model (EBM) Training

---

   % Loop over all domain pairs
3: **for** $i \in \{1,\dots,n\}; j \in \{1,\dots,n\}; i \neq j$ **do**
4:   **repeat**
5:     Sample $\mathbf{x} \sim \mathcal{D}_j$, $\mathbf{x}_0 \sim \mathcal{D}_i$.
      % Perform Langevin Dynamics
6:     **for** $k = 1,\dots,K$ **do**
7:       Sample $\epsilon \sim \mathcal{N}(0, I)$.
8:       $\mathbf{x}_k = \mathbf{x}_{k-1} - \frac{\beta^2}{2}\nabla_{\mathbf{x}_{k-1}}E_{\theta_{ij}}(\mathbf{x}_{k-1}) + \beta\epsilon$.
9:     **end for**
10:     $\tilde{\mathbf{x}} \leftarrow \mathbf{x}_K$.
      % Compute gradient
11:     $\nabla_{\theta_{ij}}\mathcal{L} = \nabla_{\theta_{ij}}E_{\theta_{ij}}(\mathbf{x}) - \nabla_{\theta_{ij}}E_{\theta_{ij}}(\tilde{\mathbf{x}})$.
12:     $\theta_{ij} \leftarrow \theta_{ij} - \lambda\nabla_{\theta_{ij}}\mathcal{L}$.
13:   **until** convergence
14: **end for**

---

15: Generation of Langevin Samples

---

   % Storage for augmented datasets
16: Initialize empty sets $\{\mathcal{D}_{ij}^k\}_{i=1,j=1,k=1}^{n,n,K}$.
   % Loop over all domain pairs
17: **for** $i \in \{1,\dots,n\}; j \in \{1,\dots,n\}; i \neq j$ **do**
17:   % Loop through samples in source domain $i$
18:   **for** $\mathbf{x}_i \in \mathcal{D}_i$ **do**
19:     $\mathbf{x}_0 \leftarrow \mathbf{x}_i$.
      % Perform Langevin steps
20:     **for** $k = 1,\dots,K$ **do**
21:       Sample $\epsilon \sim \mathcal{N}(0, I)$.
22:       $\mathbf{x}_k = \mathbf{x}_{k-1} - \frac{\beta^2}{2}\nabla_{\mathbf{x}_{k-1}}E_{\theta_{ij}}(\mathbf{x}_{k-1}) + \beta\epsilon$.
      % Store augmented sample
23:       $(\mathcal{D}_{ij}^k)$.append($\mathbf{x}_k$).
24:     **end for**
25:   **end for**
26: **end for**
   % Return all augmented datasets
27: **return** $\{\mathcal{D}_{ij}^k\}_{i=1,j=1,k=1}^{n,n,K}$.

---

more efficient than FedDG (4.60 hrs), while providing superior performance.

Additionally, the average Energy-Based Model (EBM) training time is approximately 0.357 hours per source-target domain pair. The inference cost of running a Langevin

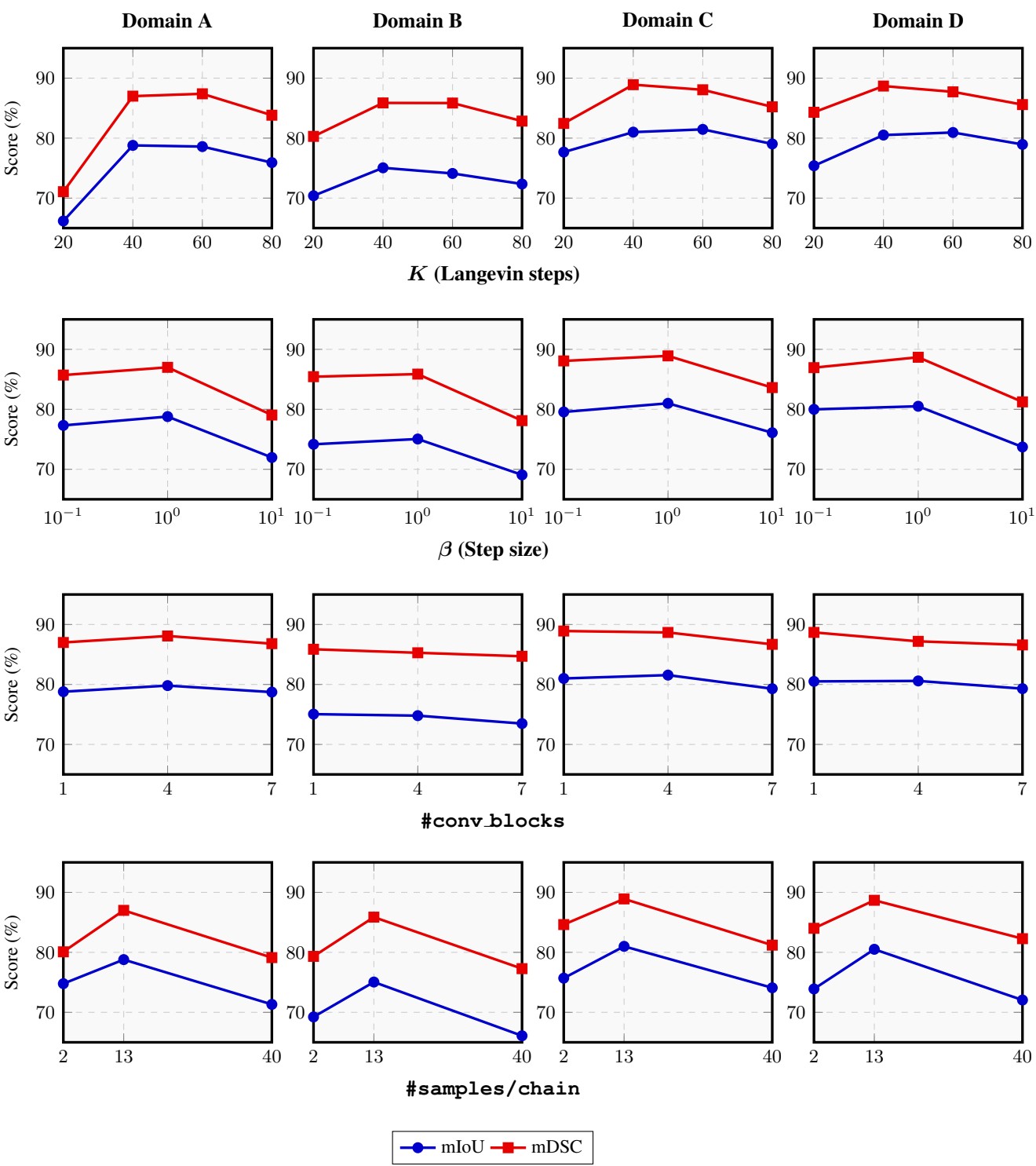

Figure 5: Ablation analysis across four domains. The first row shows the effect of varying the number of Langevin steps ($k$); the second row shows the effect of varying the step size ($\beta$); the third row depicts the impact of changing the number of convolutional blocks; and the fourth row presents the effect of varying the number of Langevin samples per chain. Metrics reported are mIoU and mDSC.

Table 5: Training time (in GPU hours) and peak memory usage (in GB) for all methods. "+Ours" refers to integration of LangDAug into the baseline.

| Metric | ERM | Ours | FedDG | FedDG+Ours | RAM | RAM+Ours | TriD | TriD+Ours |
|---|---|---|---|---|---|---|---|---|
| GPU hrs | 1.51 | 3.14 | 4.60 | 6.13 | 2.75 | 3.77 | 5.53 | 7.49 |
| Memory (GB) | 10.36 | 19.41 | 16.77 | 23.16 | 12.58 | 20.24 | 24.87 | 30.11 |

Dynamics (LD) chain is minimal, taking roughly 2 seconds.

Apart from this, the proposed method also requires additional storage requirements to store the Langevin samples. Particularly, for $n$ data points, and a saving frequency of $f$, the additional data points will equal $nK/f$.

As with many domain augmentation (DA) methods that rely on synthetic sample generation, increased training cost is an inherent trade-off. Potential directions for optimization include selective sampling strategies (e.g., coresets) and architectural sharing for EBMs to enable conditioning across domains.

## D. Inter-Domain Transversal Visualizations

We present visual examples of the inter-domain transversal process over $K = 40$ Langevin dynamics steps. The transversal sequences for the retinal fundus dataset are depicted in Figures 6 to 9, while those for the prostate dataset are shown in Figures 10 to 15. In each figure, rows correspond to distinct samples starting from the source domain, with each row capturing the intermediate Langevin samples, while columns represent the specific steps in the Langevin Dynamics process at which the transversal are recorded.

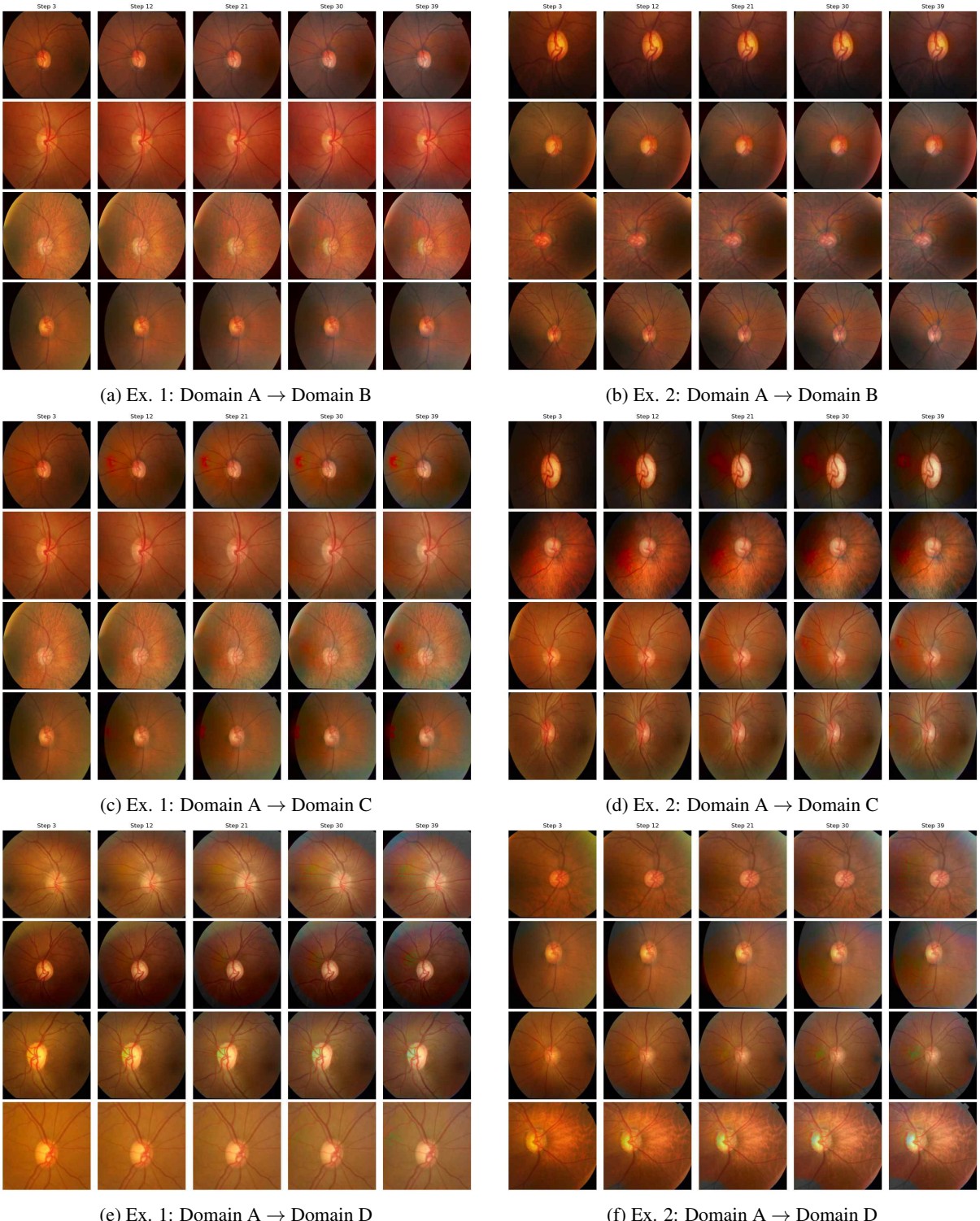

(a) Ex. 1: Domain A → Domain B

(b) Ex. 2: Domain A → Domain B

(c) Ex. 1: Domain A → Domain C

(d) Ex. 2: Domain A → Domain C

(e) Ex. 1: Domain A → Domain D

(f) Ex. 2: Domain A → Domain D

Figure 6: Examples of translation from Domain A to Domains B, C and D using the proposed method on the retinal fundus dataset.

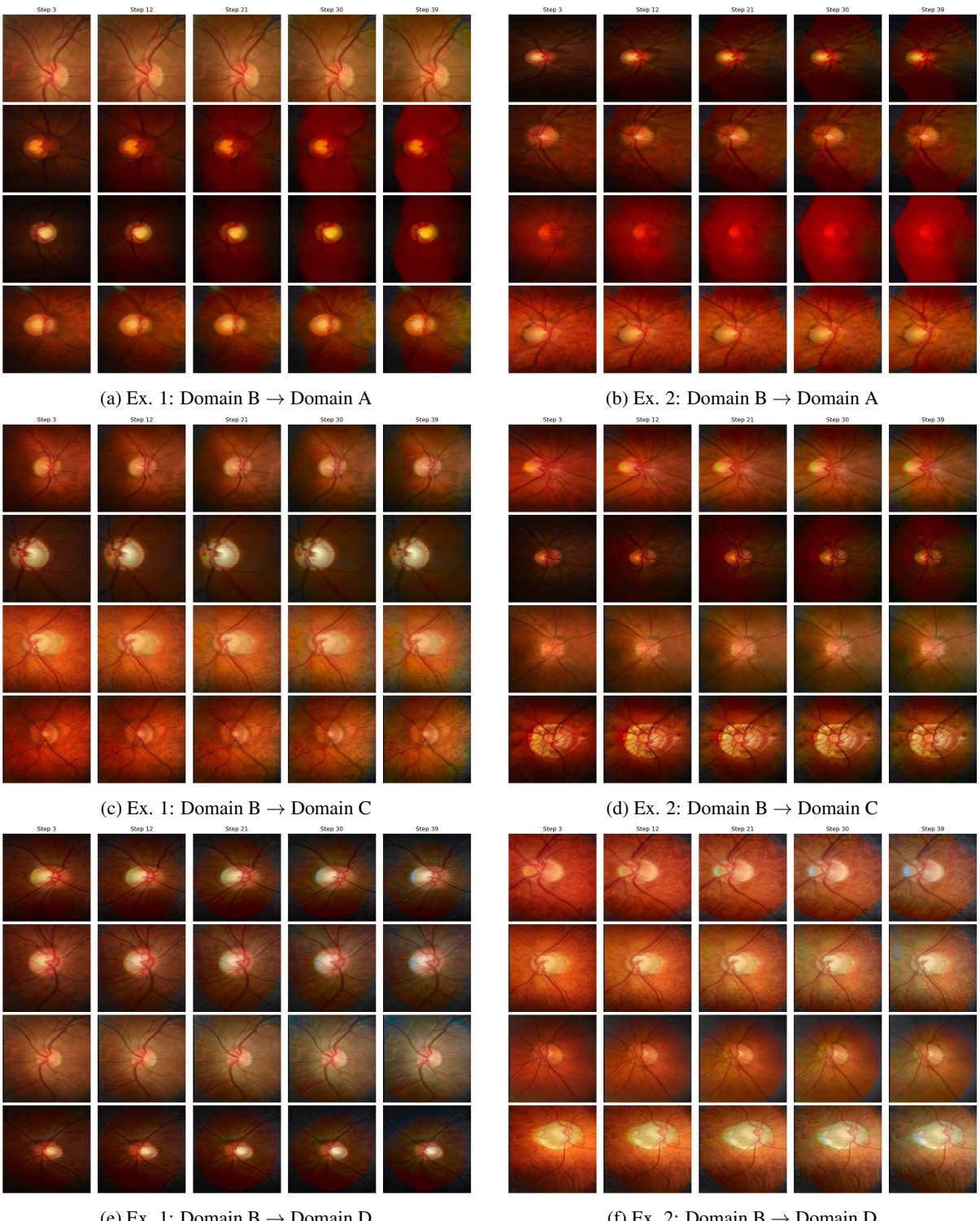

(a) Ex. 1: Domain B → Domain A

(b) Ex. 2: Domain B → Domain A

(c) Ex. 1: Domain B → Domain C

(d) Ex. 2: Domain B → Domain C

(e) Ex. 1: Domain B → Domain D

(f) Ex. 2: Domain B → Domain D

Figure 7: Examples of translation from Domain B to Domains A, C and D using the proposed method on the retinal fundus dataset.

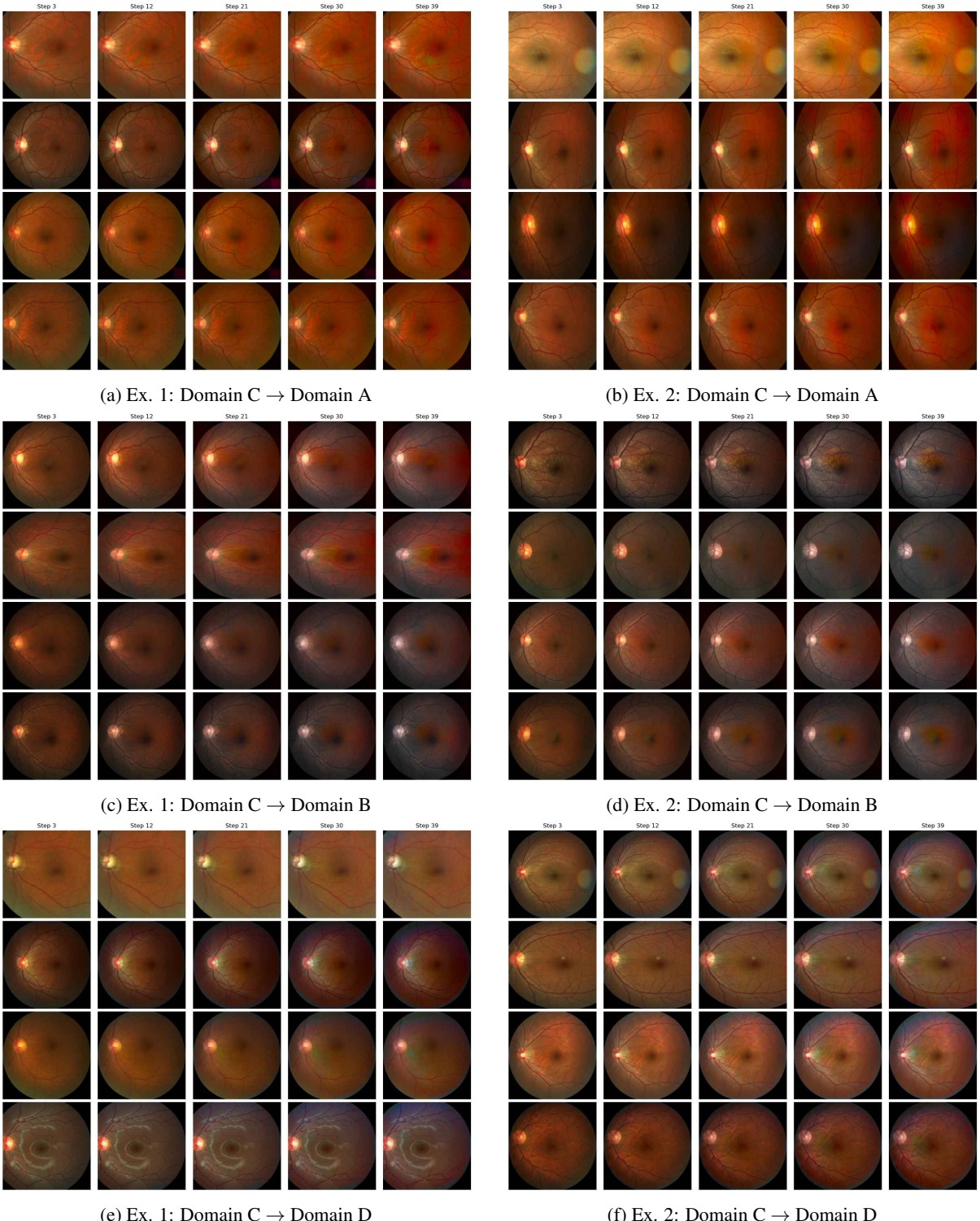

(a) Ex. 1: Domain C → Domain A

(b) Ex. 2: Domain C → Domain A

(c) Ex. 1: Domain C → Domain B

(d) Ex. 2: Domain C → Domain B

(e) Ex. 1: Domain C → Domain D

(f) Ex. 2: Domain C → Domain D

Figure 8: Examples of translation from Domain C to Domains A, B and D using the proposed method on the retinal fundus dataset.

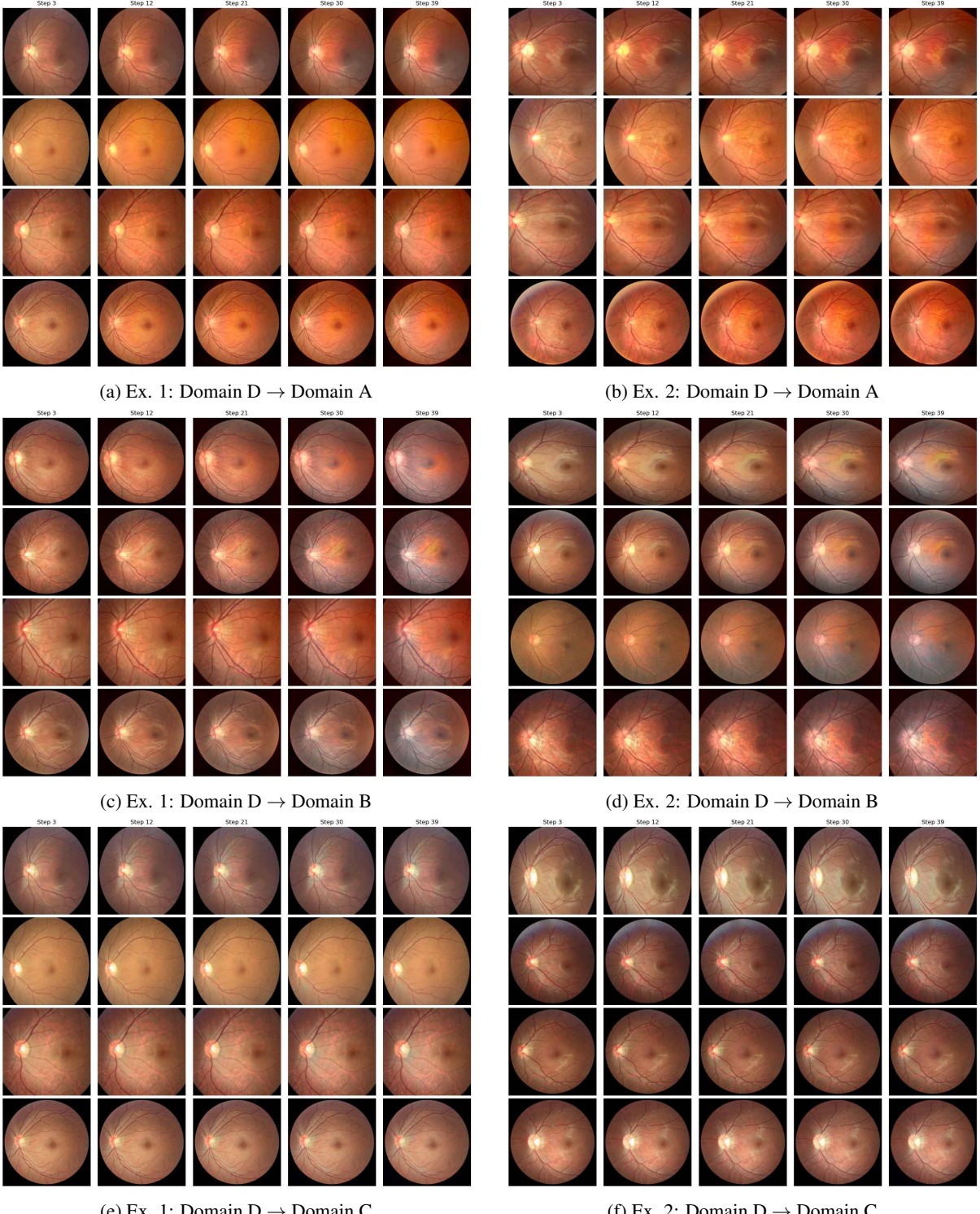

(a) Ex. 1: Domain D → Domain A

(b) Ex. 2: Domain D → Domain A

(c) Ex. 1: Domain D → Domain B

(d) Ex. 2: Domain D → Domain B

(e) Ex. 1: Domain D → Domain C

(f) Ex. 2: Domain D → Domain C

Figure 9: Examples of translation from Domain D to Domains A, B and C using the proposed method on the retinal fundus dataset.

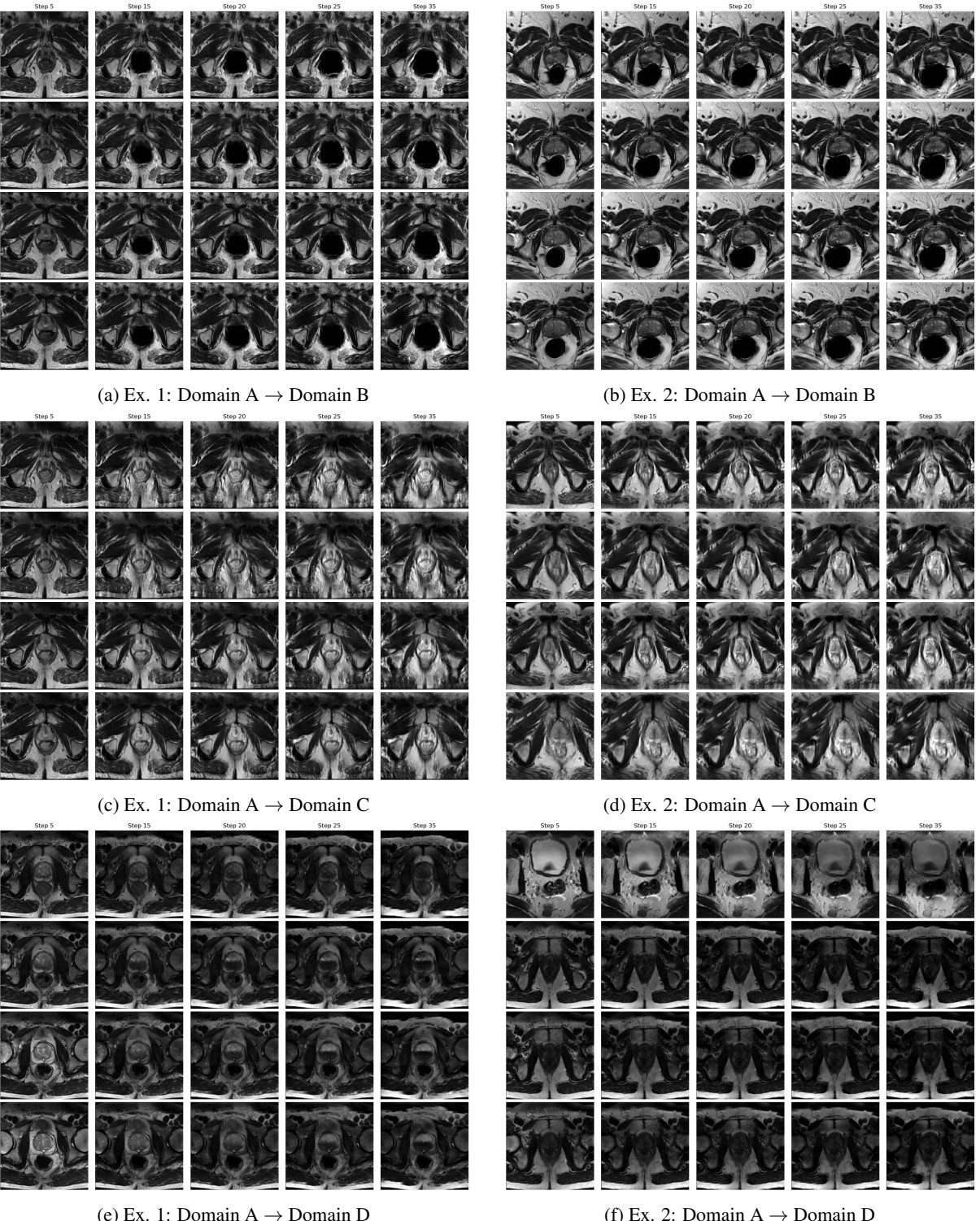

(a) Ex. 1: Domain A → Domain B

(b) Ex. 2: Domain A → Domain B

(c) Ex. 1: Domain A → Domain C

(d) Ex. 2: Domain A → Domain C

(e) Ex. 1: Domain A → Domain D

(f) Ex. 2: Domain A → Domain D

Figure 10: Examples of translation from Domain A to Domains B, C and D using the proposed method on the prostate dataset.

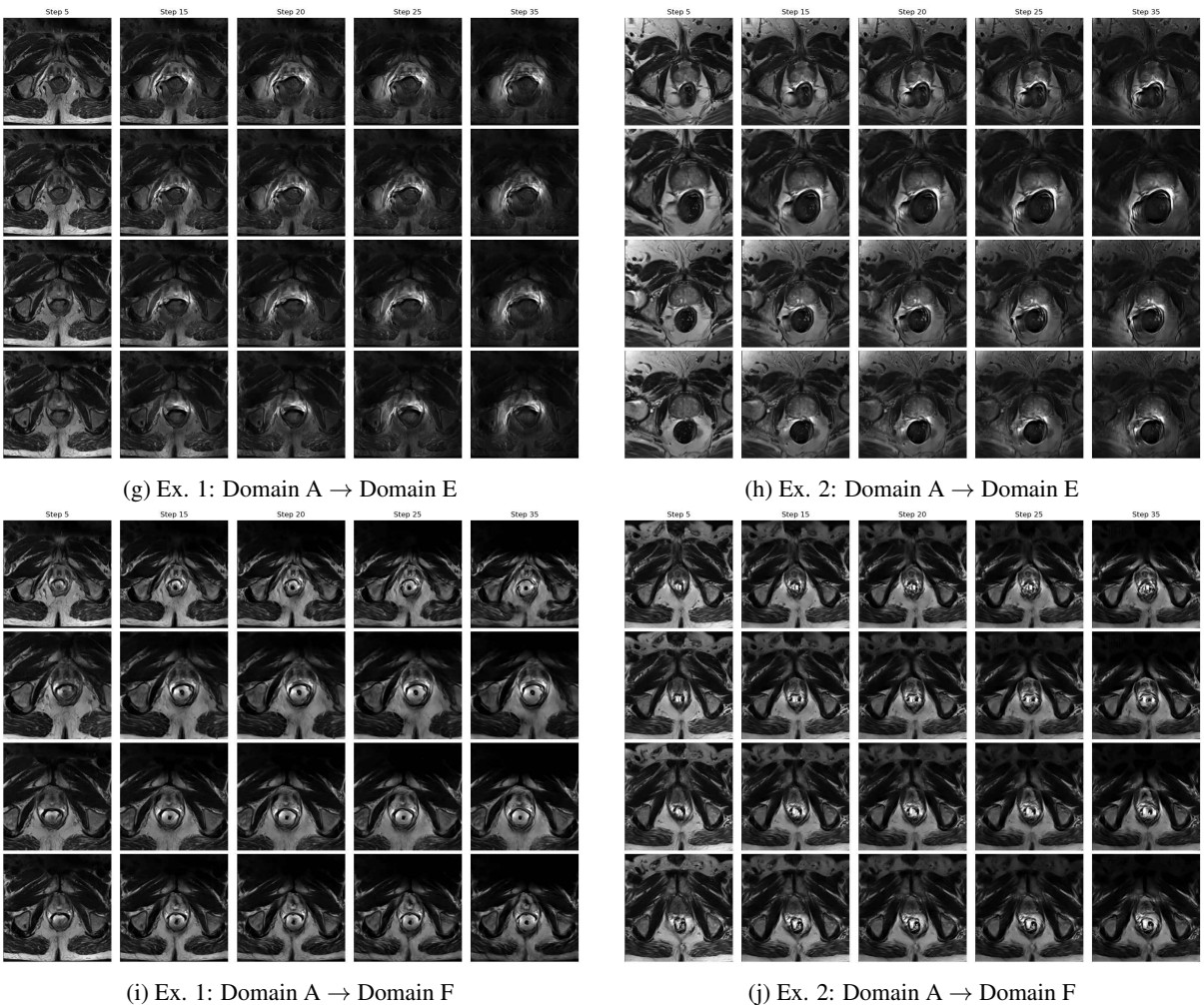

(g) Ex. 1: Domain A → Domain E        (h) Ex. 2: Domain A → Domain E

(i) Ex. 1: Domain A → Domain F        (j) Ex. 2: Domain A → Domain F

Figure 10: Examples of translation from Domain A to Domains E and F using the proposed method on the prostate dataset.

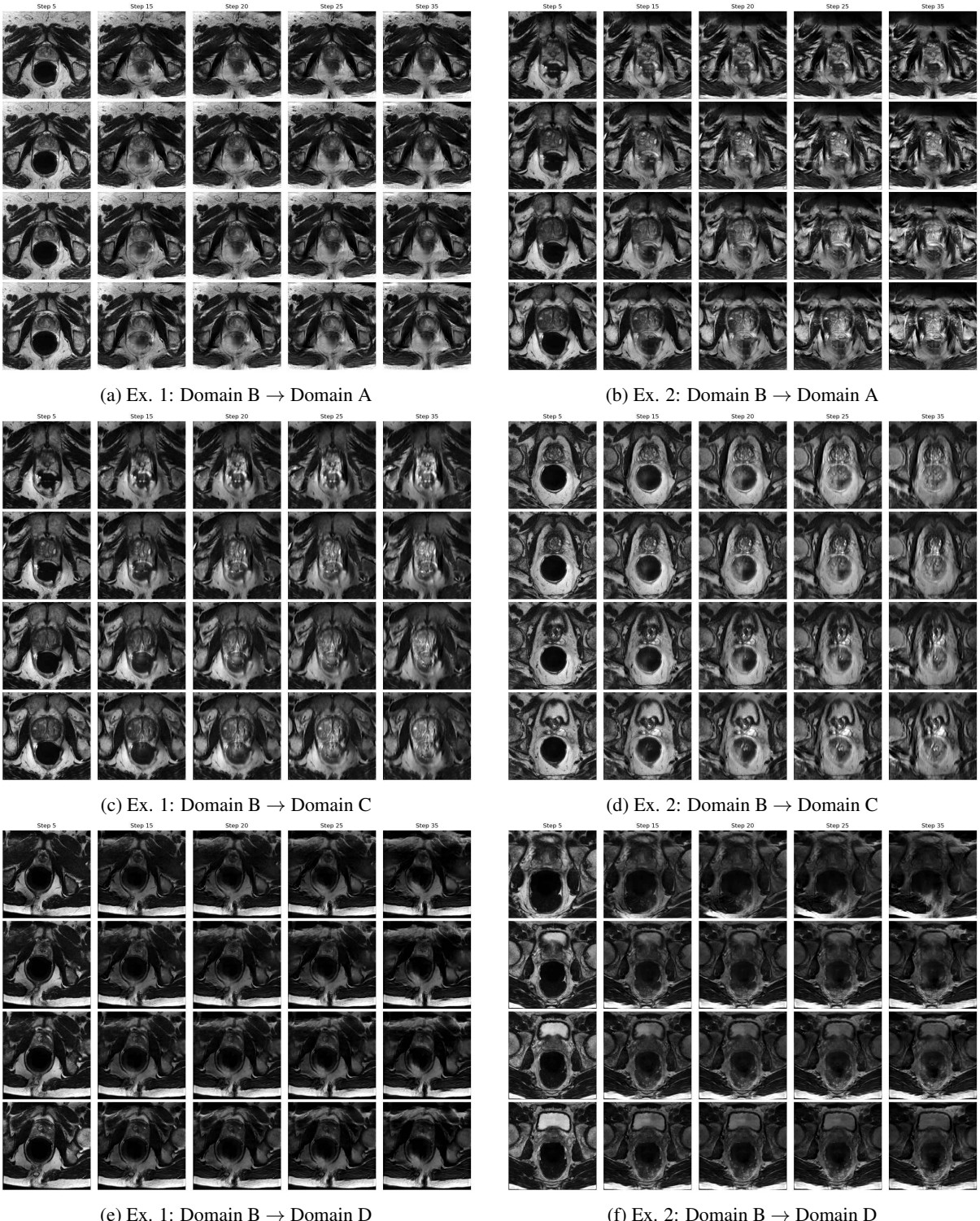

(a) Ex. 1: Domain B → Domain A
(b) Ex. 2: Domain B → Domain A

(c) Ex. 1: Domain B → Domain C
(d) Ex. 2: Domain B → Domain C

(e) Ex. 1: Domain B → Domain D
(f) Ex. 2: Domain B → Domain D

Figure 11: Examples of translation from Domain B to Domains A, C and D using the proposed method on the prostate dataset.

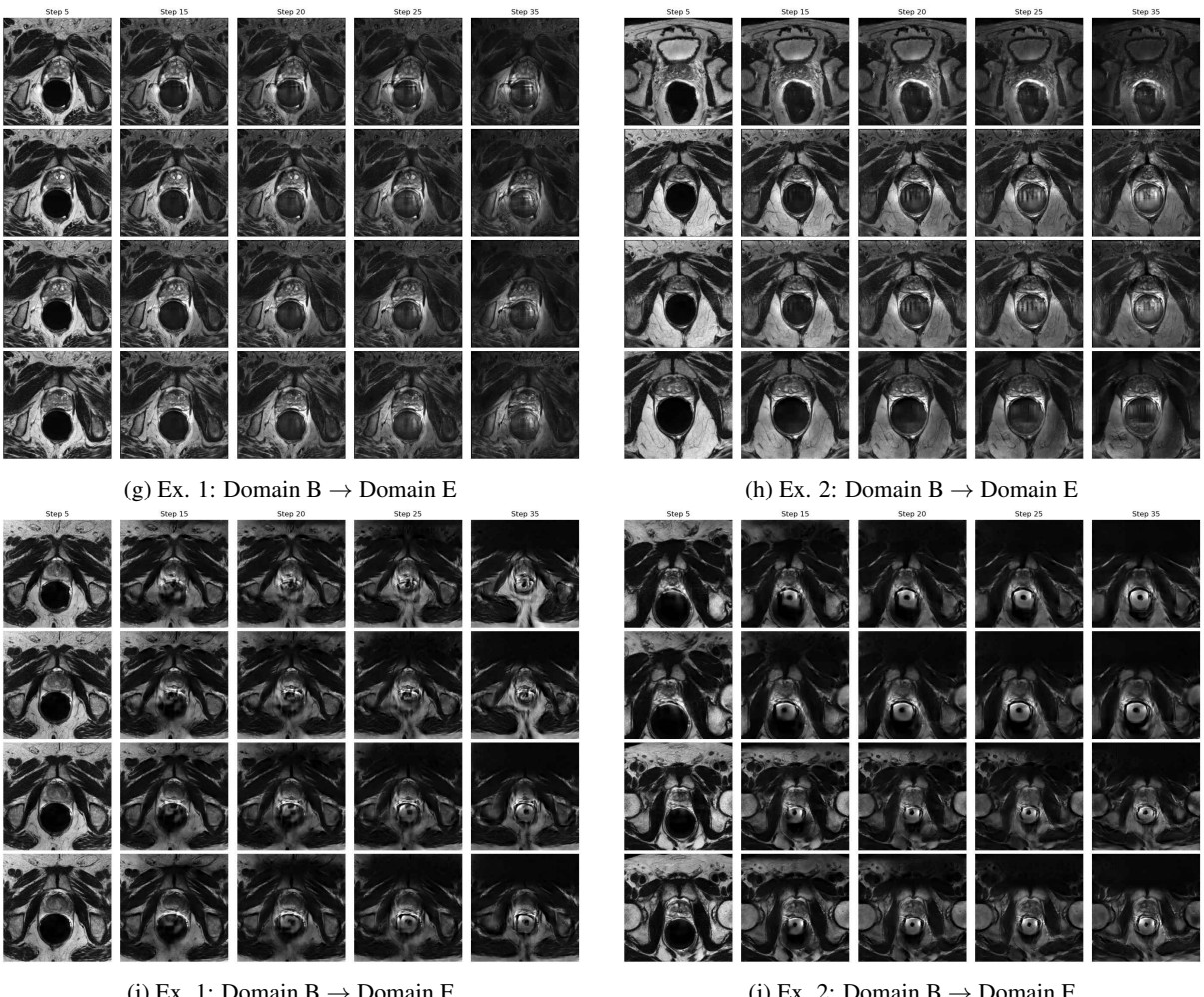

(g) Ex. 1: Domain B → Domain E          (h) Ex. 2: Domain B → Domain E

(i) Ex. 1: Domain B → Domain F          (j) Ex. 2: Domain B → Domain F

Figure 11: Examples of translation from Domain B to Domains E and F using the proposed method on the prostate dataset.

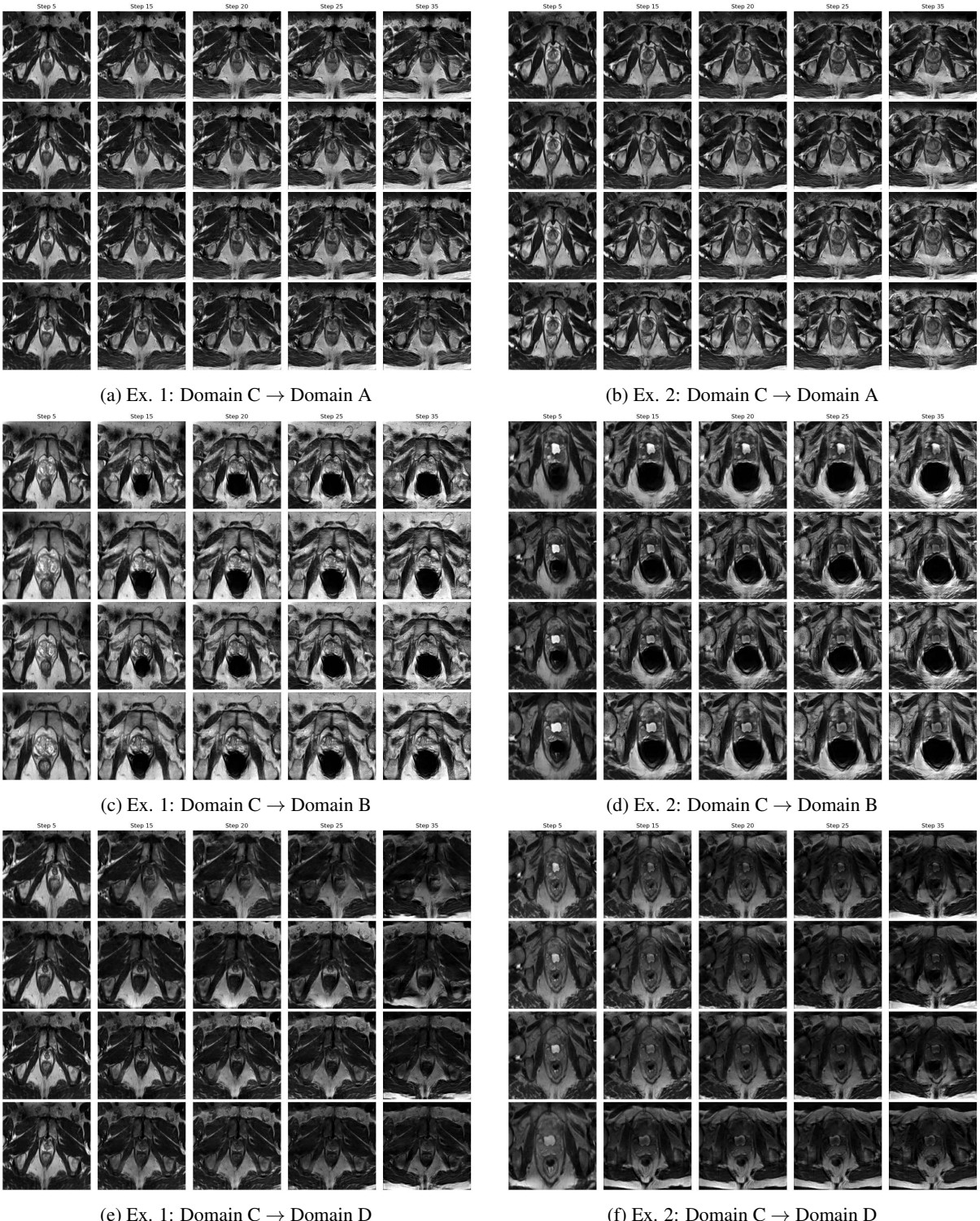

(a) Ex. 1: Domain C → Domain A

(b) Ex. 2: Domain C → Domain A

(c) Ex. 1: Domain C → Domain B

(d) Ex. 2: Domain C → Domain B

(e) Ex. 1: Domain C → Domain D

(f) Ex. 2: Domain C → Domain D

Figure 12: Examples of translation from Domain C to Domains A, B, and D using the proposed method on the prostate dataset.

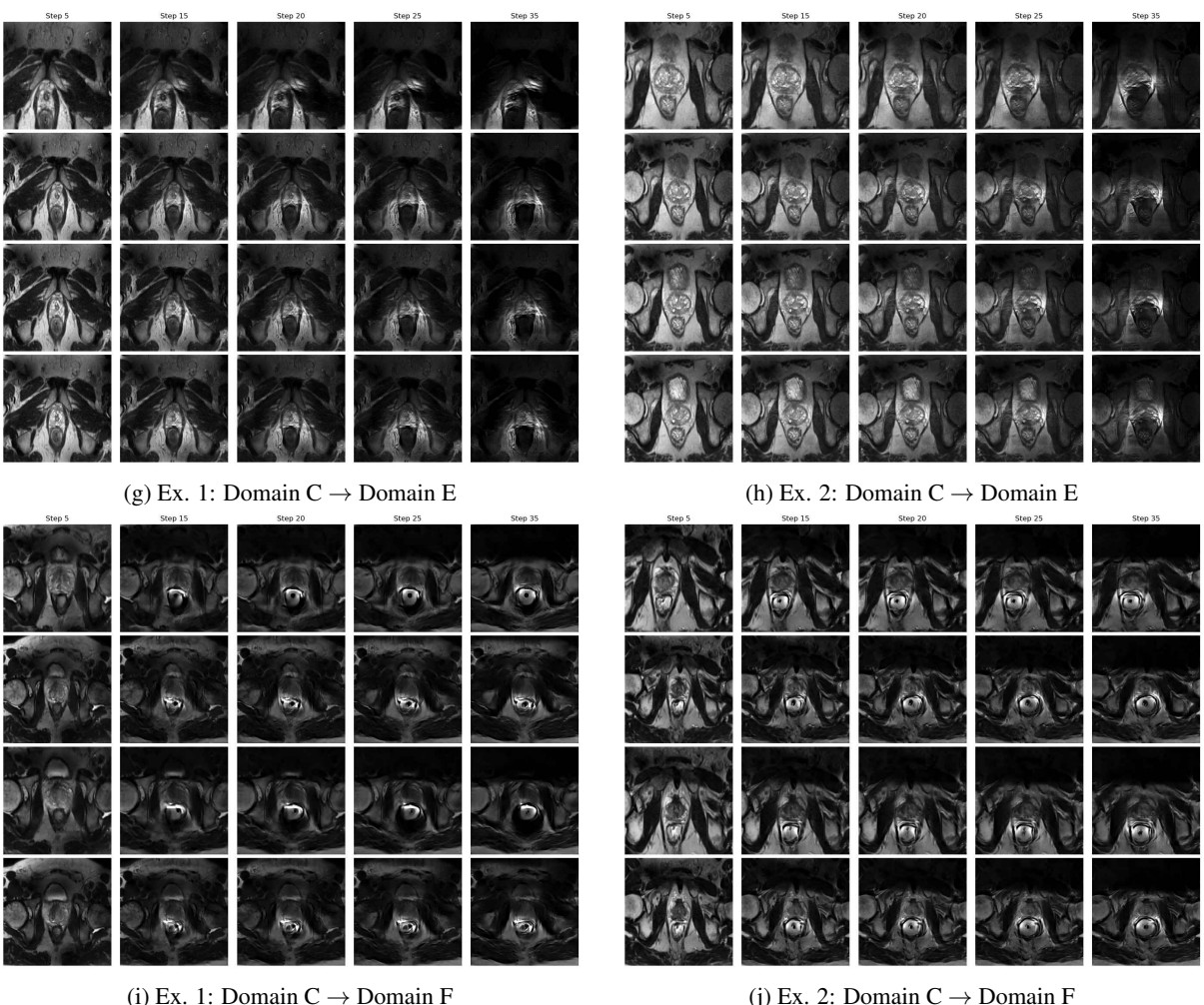

(g) Ex. 1: Domain C → Domain E     (h) Ex. 2: Domain C → Domain E

(i) Ex. 1: Domain C → Domain F     (j) Ex. 2: Domain C → Domain F

Figure 12: Examples of translation from Domain C to Domains E and F using the proposed method on the prostate dataset.

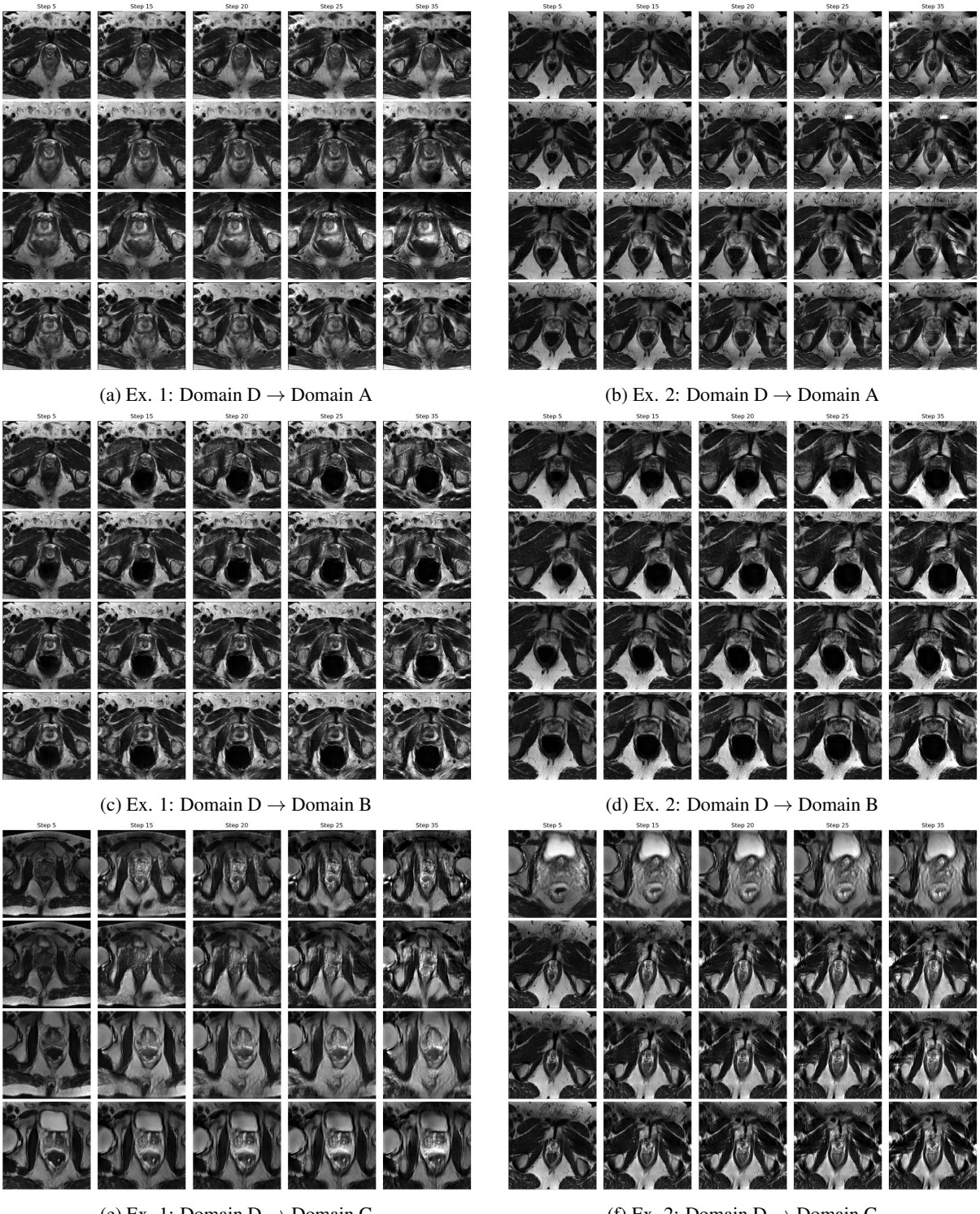

(a) Ex. 1: Domain D → Domain A    (b) Ex. 2: Domain D → Domain A

(c) Ex. 1: Domain D → Domain B    (d) Ex. 2: Domain D → Domain B

(e) Ex. 1: Domain D → Domain C    (f) Ex. 2: Domain D → Domain C

Figure 13: Examples of translation from Domain D to Domains A, B and C using the proposed method on the prostate dataset.

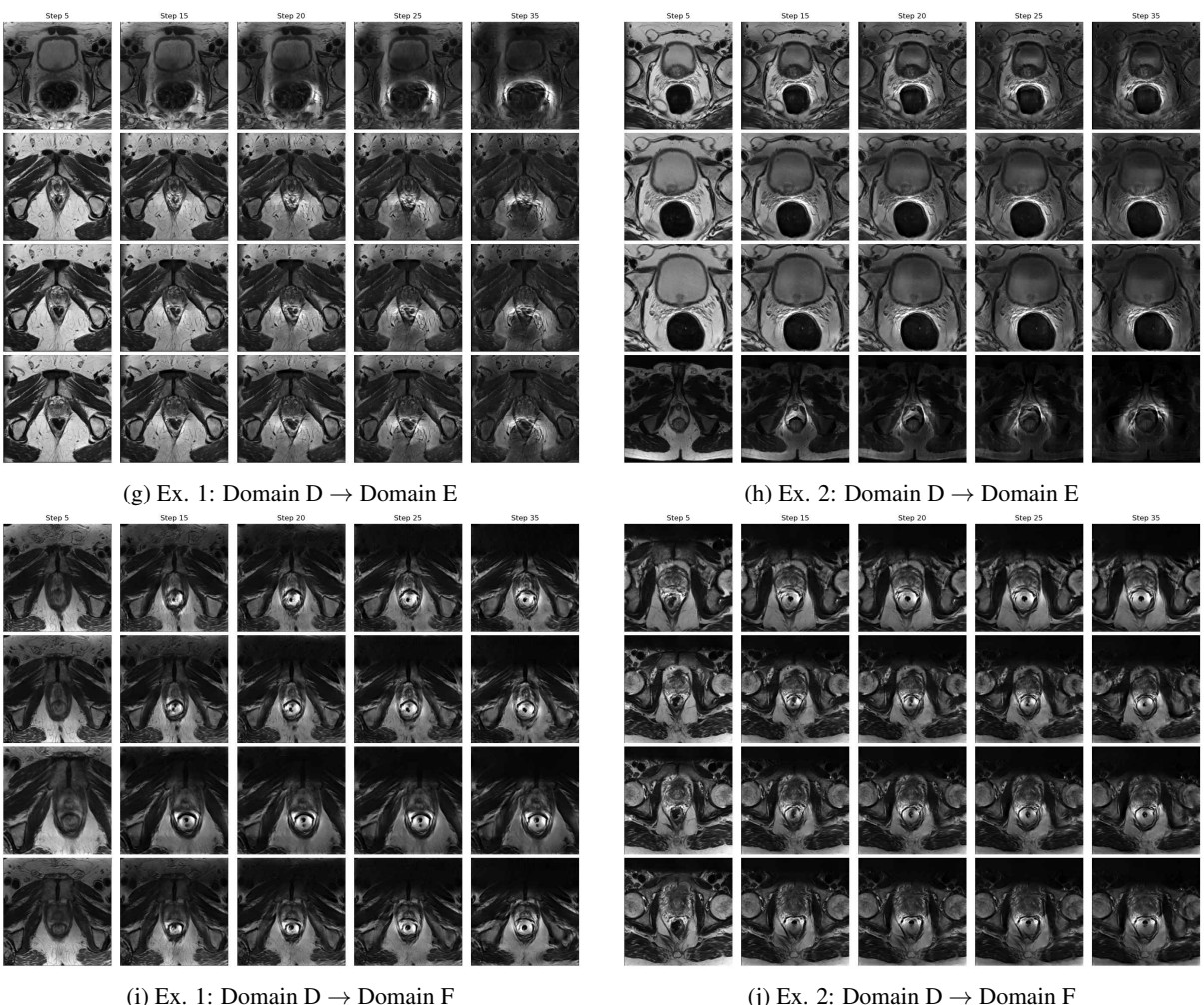

(g) Ex. 1: Domain D → Domain E

(h) Ex. 2: Domain D → Domain E

(i) Ex. 1: Domain D → Domain F

(j) Ex. 2: Domain D → Domain F

Figure 13: Examples of translation from Domain D to Domains E and F using the proposed method on the prostate dataset.

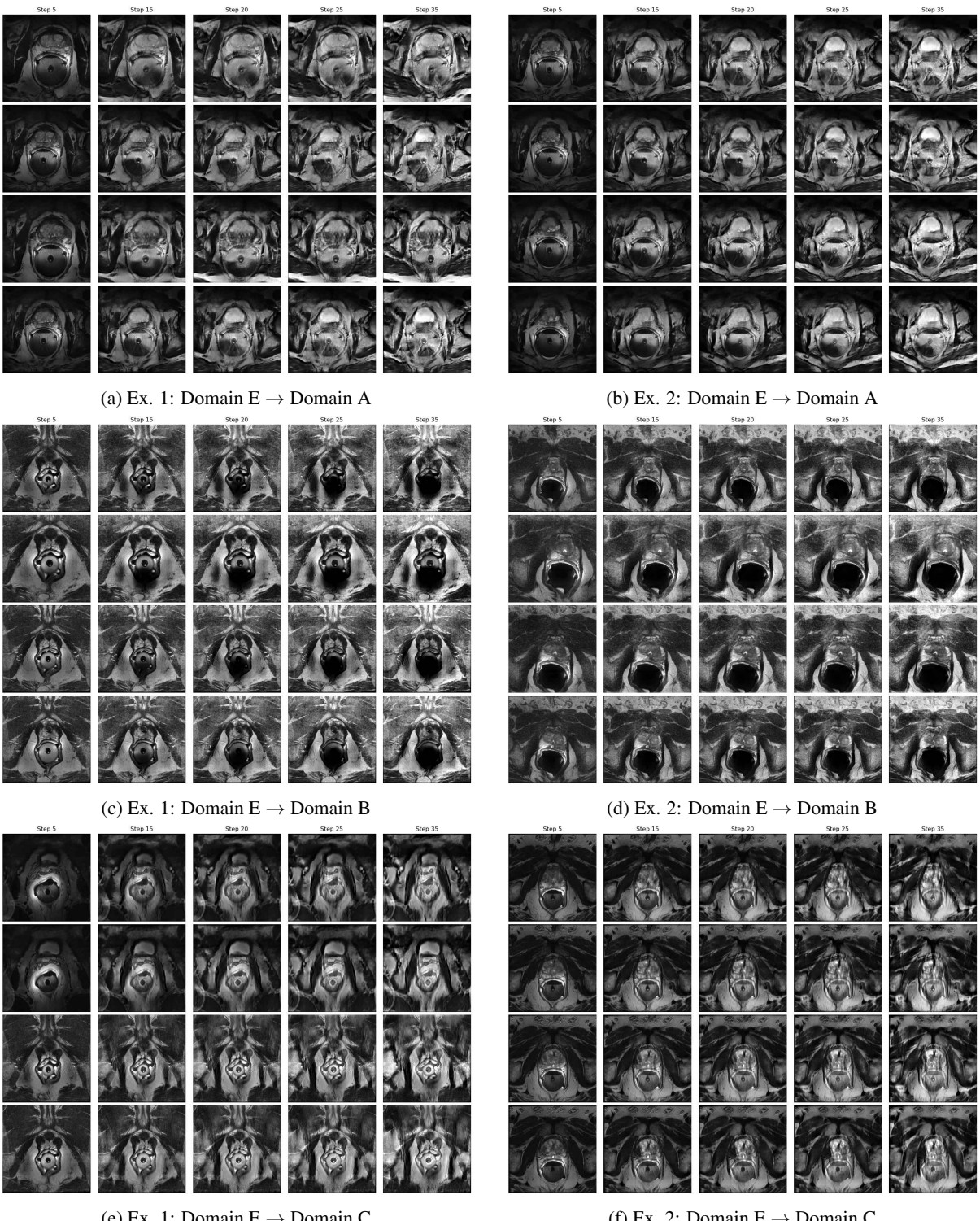

(a) Ex. 1: Domain E → Domain A    (b) Ex. 2: Domain E → Domain A

(c) Ex. 1: Domain E → Domain B    (d) Ex. 2: Domain E → Domain B

(e) Ex. 1: Domain E → Domain C    (f) Ex. 2: Domain E → Domain C

Figure 14: Examples of translation from Domain E to Domains A, B and C using the proposed method on the prostate dataset.

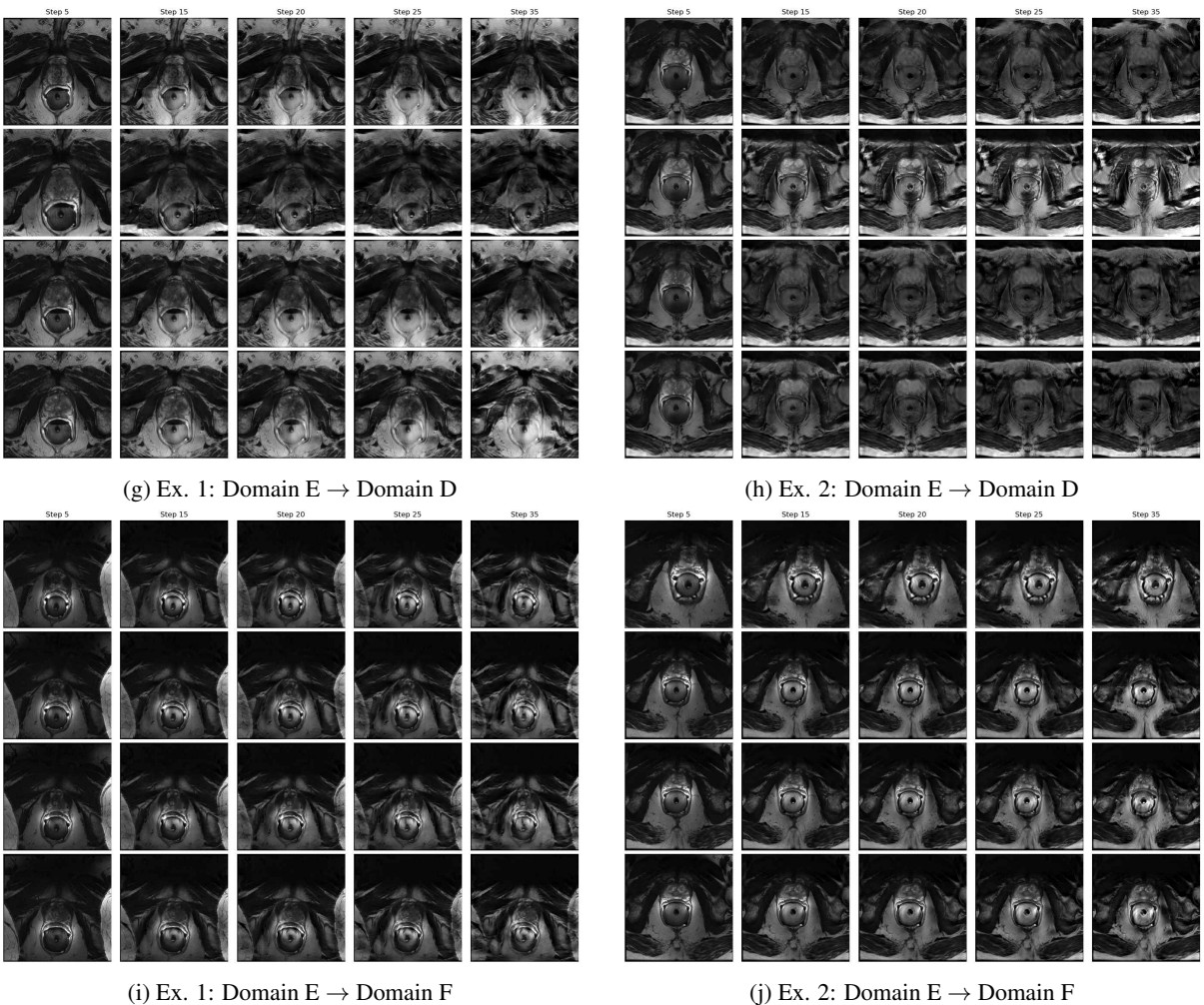

(g) Ex. 1: Domain E → Domain D

(h) Ex. 2: Domain E → Domain D

(i) Ex. 1: Domain E → Domain F

(j) Ex. 2: Domain E → Domain F

Figure 14: Examples of translation from Domain E to Domains D and F using the proposed method on the prostate dataset.

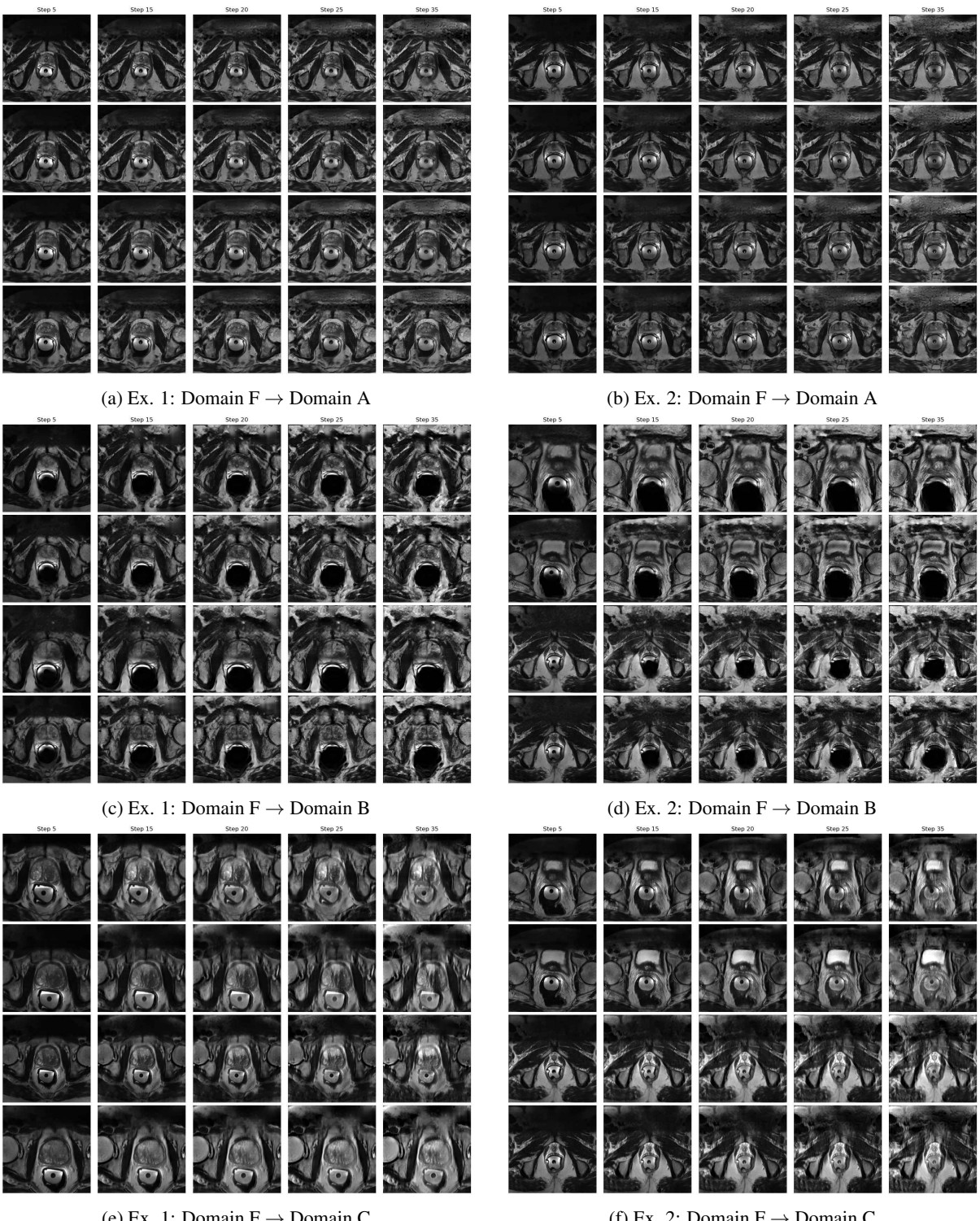

(a) Ex. 1: Domain F → Domain A       (b) Ex. 2: Domain F → Domain A

(c) Ex. 1: Domain F → Domain B       (d) Ex. 2: Domain F → Domain B

(e) Ex. 1: Domain F → Domain C       (f) Ex. 2: Domain F → Domain C

Figure 15: Examples of translation from Domain F to Domains A, B and C using the proposed method on the prostate dataset.

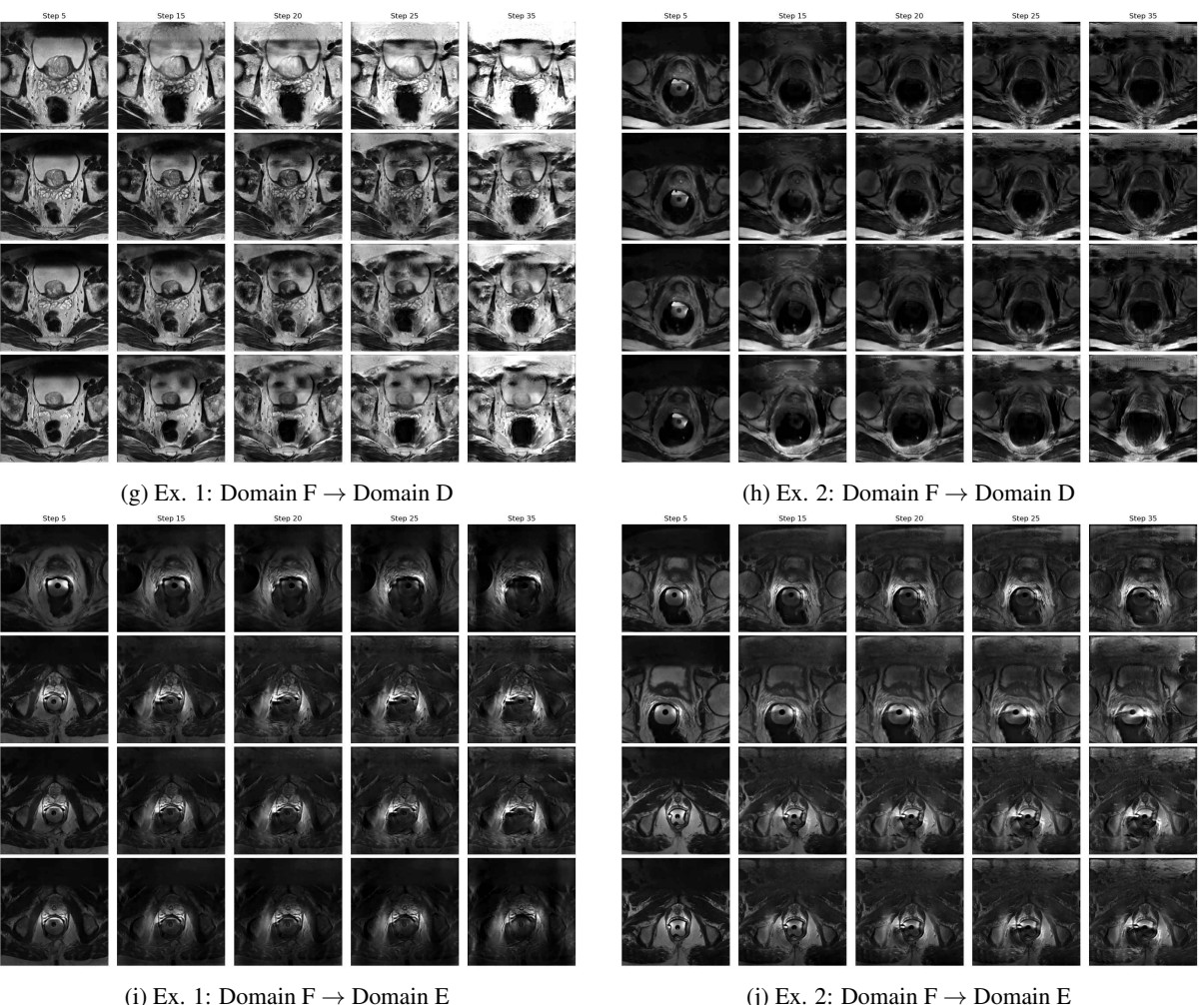

(g) Ex. 1: Domain F → Domain D       (h) Ex. 2: Domain F → Domain D

(i) Ex. 1: Domain F → Domain E       (j) Ex. 2: Domain F → Domain E

Figure 15: Examples of translation from Domain F to Domains D and E using the proposed method on the prostate dataset.

