# OpenReview forum: "LangDAug: Langevin Data Augmentation for Multi-Source Domain Generalization in Medical Image Segmentation"
_ICML.cc/2025/Conference — ICML 2025 poster_

### Official Review · Reviewer_WZCC · 2025-03-11

**Overall Recommendation:** 2

**Summary:**

This paper presents LangDAug, a novel data augmentation method designed to address multi-source domain generalization challenges in medical image segmentation. Leveraging Langevin Dynamics (LD) and Energy-Based Models (EBMs), LangDAug generates intermediate samples bridging source domains to enhance model generalization across unseen target domains. The approach involves training EBMs to capture energy distributions between source domains, using LD to create bridging samples, and incorporating these samples into the segmentation model's training process. Theoretical analysis demonstrates that LangDAug induces regularization, reducing Rademacher complexity and improving generalization. Experiments on retinal fundus segmentation and 2D MRI prostate segmentation datasets show that LangDAug outperforms existing domain generalization methods and effectively complements domain randomization techniques like FedDG and RAM.

## update after rebuttal
While this work is application-oriented, after thoroughly reviewing all the responses, I remain concerned that the high computational cost and storage requirement could significantly limit the practical applicability of the proposed method. Therefore, I maintain my original score as 2.

**Claims And Evidence:**

N/A

**Essential References Not Discussed:**

N/A

**Experimental Designs Or Analyses:**

N/A

**Methods And Evaluation Criteria:**

N/A

**Other Comments Or Suggestions:**

N/A

**Other Strengths And Weaknesses:**

**Strengths:**

1. LangDAug introduces a new data augmentation method combining Langevin Dynamics and Energy-Based Models, offering a new perspective for solving multi-source domain generalization problems in medical image segmentation.
2. The method is supported by theoretical analysis, demonstrating its regularization effect and its ability to reduce Rademacher complexity, enhancing the reliability and scientific foundation of the approach.
3. This method can be used independently or integrated with existing domain randomization techniques (such as FedDG and RAM), showing its flexibility and practical applicability.

**Weaknesses:**

1. Training an EBM for every pair of source domains results in significant computational and storage requirements, especially when the number of source domains increases. Generating a large number of intermediate samples substantially increases training time, which may hinder the method's practicality for large-scale datasets.
2. The effectiveness of LangDAug heavily relies on the quality of the trained EBMs, but the paper lacks an in-depth discussion of EBM training stability and its potential impact on results.
3. Generating inter-domain samples to improve generalization is a reliable solution. However, this requires that the samples in the training set cover the sample space uniformly to obtain good interpolation results. What happens to the performance of the model if the distribution of samples in the training set is restricted?

**Questions For Authors:**

Pls refer to Weaknesses

**Relation To Broader Scientific Literature:**

N/A

**Theoretical Claims:**

N/A

---

> ### Author Rebuttal · Authors · 2025-03-30
>
> Thanks for your comments. We have tried our best to answer your concerns. We will be happy to engage in follow-up discussion if you have further questions.
>
> ## Computational and Storage Requirements
> Thanks for this comment. We acknowledged this in our limitations, our method increases storage and computational costs.
> -  Additional storage is needed for Langevin samples - specifically $(k/f)×n$ samples per domain pair (where $f$ is saving frequency, $n$ is total data points).
> - Even though we train EBMs for each pair, our EBMs are lightweight and require only 0.357 hr/pair for training. Further, this can be parallelized to reduce this time.
> - EBM training is performed offline, and models can be discarded after generating samples. This doesn't affect the main segmentation network's training or inference requirements.
> - As shown in response to reviewer 1jQ9, LangDAug's training time (3.14 hrs) are comparable to existing methods like RAM (2.75 hrs) and less than FedDG (4.60 hrs).
> - Increased training cost is an inherent limitation of DA methods that use synthetic samples. Solutions could include selective sampling (e.g., coresets) and shared architectures to jointly train EBMs with conditioning on source/target domains.We leave these techniques to be explored in future works.
> ## Stability of EBMs
> We appreciate the reviewer’s comment regarding the importance of EBM training stability in performance. While our submission mentioned training details, we didn't observe any instability. To show this, we conduct additional ablations to assess the stability of the EBMs used in our work. Specifically, we monitor the segmentation performance of our method against different EBM hparam settings. We observe the effect of number of langevin steps ($k$), step size ($\beta$) and EBM complexity (by varying the number of conv blocks):
> |                         | Domain 1 |       | Domain 2 |       | Domain 3 |       | Domain 4 |       |
> |-------------------------|----------|-------|----------|-------|----------|-------|----------|-------|
> |                         | mIoU     | mDSC  | mIoU     | mDSC  | mIoU     | mDSC  | mIoU     | mDSC  |
> | $k$                     |          |       |          |       |          |       |          |       |
> | 20                      | 66.18    | 71.07 | 70.39    | 80.29 | 77.67    | 82.45 | 75.39    | 84.31 |
> | 40                      | 78.79    | 87.00 | 75.05    | 85.87 | 81.00    | 88.91 | 80.51    | 88.68 |
> | 60                      | 78.59    | 87.39 | 74.11    | 85.85 | 81.46    | 88.06 | 80.94    | 87.72 |
> | 80                      | 75.92    | 83.82 | 72.35    | 82.84 | 79.03    | 85.22 | 78.95    | 85.59 |
> | $\beta$                 |          |       |          |       |          |       |          |       |
> | 0.1                     | 77.32    | 85.71 | 74.17    | 85.44 | 79.56    | 88.07 | 79.99    | 86.95 |
> | 1                       | 78.79    | 87.00 | 75.05    | 85.87 | 81.00    | 88.91 | 80.51    | 88.68 |
> | 10                      | 71.97    | 79.05 | 69.07    | 78.11 | 76.10    | 83.62 | 73.72    | 81.24 |
> | #Conv blocks/resolution |          |       |          |       |          |       |          |       |
> | 1                       | 78.79    | 87.00 | 75.05    | 85.87 | 81.00    | 88.91 | 80.51    | 88.68 |
> | 4                       | 79.81    | 88.08 | 74.80    | 85.28 | 81.56    | 88.67 | 80.59    | 87.19 |
> | 7                       | 78.725   | 86.80 | 73.48    | 84.70 | 79.29    | 86.69 | 79.30    | 86.59 |
>
> We see that under reasonable hparam values, the performance of LangDAug is stable. Particularly, smaller $\beta$, moderate $k$ lead to good performance. Whereas, the EBM complexity has very minimal effect on performance, indicating lightweight EBMs are sufficient. From this, we conclude stable performance of LangDAug under reasonable hparam settings.
> ## Restricted Training distribution
> Thanks for this comment. In general, all DG methods rely on diversity in source domains, as noted in Sec. 1, 2.
> This aligns with a fundamental assumption in DG - diverse source domains allow the model to learn representations for better generalization.
> When the original distribution is restricted or sparse in certain regions, LangDAug helps fill these gaps, enriching the training distribution. While extremely limited diversity would challenge any DG method, our experiments show competitive performance even in difficult cases. We refer to Tab. 1 & 2 where Domain B (Tab. 1) & Domain C (Tab. 2) are the most difficult domains because of significantly different intstruments used for image acquisition. In this case, the source domains are ineffective in capturing the corresponding target domain distribution. Despite this challenge, LangDAug outperforms other methods in several metrics for these domains. This demonstrates that while limited diversity in source domains is a fundamental challenge for all DG methods, our approach remains effective even under these constrained conditions.

---

### Official Review · Reviewer_1jQ9 · 2025-03-11

**Overall Recommendation:** 3

**Summary:**

The authors propose LangDAug, a Langevin Data Augmentation technique to improve multi-source domain generalization in medical image segmentation tasks. The core idea is to train Energy-Based Models (EBMs) via contrastive divergence to model the transitions between pairs of source domains and use Langevin dynamics (LD) to generate intermediate samples that serve as augmented data bridging domain gaps. Theoretically, the authors demonstrate that this augmentation method induces a regularization effect, leading to smoother optimization landscapes and better generalization by bounding model complexity based on intrinsic data manifold dimensionality. Experimental evaluations conducted on retinal fundus and prostate MRI segmentation datasets show that LangDAug outperforms previous methods across unseen domains.

**Claims And Evidence:**

Overall, the paper’s main claims regarding the effectiveness and robustness of the proposed LangDAug approach are convincingly supported by empirical evidence (quantitative benchmarks and visualizations) and theoretical analysis. Some limitations exist concerning scalability and detailed validation of anatomical fidelity, but these do not substantially undermine the validity or clarity of the core contributions.

**Essential References Not Discussed:**

The manuscript thoroughly references relevant literature on domain generalization (DG), energy-based models (EBMs), Langevin dynamics, and medical image segmentation.

**Experimental Designs Or Analyses:**

The experimental design and analyses used in this submission are sound, valid, and appropriately rigorous for assessing domain generalization performance.

However, there are still some limitations:
- Missing detailed computational cost analyses as the proposed LangDAug could increase training time.
- It is better to analyze the contribution of individual components of LangDAug (e.g., varying Langevin steps, EBMs complexity, or the number of augmented samples).
- How about applying LangDAug to 3D medical image segmentation tasks?

**Methods And Evaluation Criteria:**

The proposed methodology and evaluation criteria used in the submission are clearly appropriate and well-justified for the stated objectives of the paper; however, I would also want to know if the proposed LangDAug could be used for 3D medical image segmentation tasks and the efficiency of doing so.

**Other Comments Or Suggestions:**

NA

**Other Strengths And Weaknesses:**

NA

**Questions For Authors:**

- Could the authors provide a detailed computational cost analysis (e.g., GPU hours, memory requirements, inference and training times) of the proposed LangDAug method compared to the baseline domain generalization approaches?

- The current implementation and validation of LangDAug focus exclusively on 2D medical image segmentation. Could the authors explain any specific technical or methodological reasons why you have not extended LangDAug directly to 3D medical image segmentation tasks? Are there fundamental challenges, such as increased complexity, computational demands, or theoretical limitations, that prevented immediate extension?

**Relation To Broader Scientific Literature:**

- The key contributions (LangDAug) of this paper closely align with and extend several ideas from the broader scientific literature, specifically related to domain generalization, energy-based models, Langevin dynamics, and data augmentation strategies in machine learning, particularly in medical imaging.

- LangDAug directly extends prior work on domain generalization, especially the notion that models trained under the Empirical Risk Minimization (ERM) paradigm often fail to generalize under domain shifts (Gulrajani & Lopez-Paz, 2021; Ganin et al., 2016). It is positioned within data manipulation approaches, specifically leveraging the concept of intermediate-domain traversal via Langevin dynamics, distinguishing itself from prior DG methods that primarily relied on style-mixing (MixStyle), random convolutions (RandConv), or frequency domain augmentation (FedDG, RAM, TriD).

**Theoretical Claims:**

Overall, the theoretical claims made by the authors are clearly articulated, logically consistent, and mathematically rigorous. The authors provide a coherent theoretical framework that convincingly justifies why LangDAug would improve domain generalization. There are no substantial mathematical errors or inconsistencies evident from the provided derivations.

---

> ### Author Rebuttal · Authors · 2025-03-30
>
> Thank you for your appreciation and in-depth review. We have tried our best to answer your concerns. We will include these in final version of the paper. Please let us know if you have any other questions.
> ### Computational Cost Analyses
> We acknowledged the increased computational cost of LangDAug in our limitations. Below, we provide details of the training time and memory usage across methods on the RFS data (averaged across domains on single 48GB A6000 card):
> | Metric | ERM | Ours | FedDG | FedDG+Ours | RAM | RAM+Ours | TriD | TriD+Ours |
> |--------|---------|------|-------|------------|-----|----------|------|-----------|
> | GPU hrs/train. time | 1.508 | 3.141 | 4.604 | 6.129 | 2.746 | 3.772 | 5.528 | 7.494 |
> | Peak Memory (GB) | 10.36 | 19.41 | 16.77 | 23.16 | 12.58 | 20.24 | 24.87 | 30.11 |
>
> While LangDAug increases training time, it's important to note that LangDAug's time (3.14 hrs) is comparable to existing methods like RAM (2.75 hrs) and less than FedDG (4.60 hrs). Moreover, LangDAug performs better than these methods.
>
> In addition to above, we note an average EBM training time of 0.357 hr/pair. Further the inference cost (to run one LD chain) is very minimal ~2 sec.
>
> Increased training cost is an inherent limitation of DA methods that use synthetic samples. Future optimizations could include selective sampling (e.g., coresets) and shared architectures to jointly train EBMs with conditioning on source/target domains.
> ## Effect of each component
> Thanks for this suggestion. We provide ablations for several components of our method in the following tables:
> |                         | Domain 1 |       | Domain 2 |       | Domain 3 |       | Domain 4 |       |
> |-------------------------|----------|-------|----------|-------|----------|-------|----------|-------|
> |                         | mIoU     | mDSC  | mIoU     | mDSC  | mIoU     | mDSC  | mIoU     | mDSC  |
> | $k$                     |          |       |          |       |          |       |          |       |
> | 20                      | 66.18    | 71.07 | 70.39    | 80.29 | 77.67    | 82.45 | 75.39    | 84.31 |
> | 40                      | 78.79    | 87.00 | 75.05    | 85.87 | 81.00    | 88.91 | 80.51    | 88.68 |
> | 60                      | 78.59    | 87.39 | 74.11    | 85.85 | 81.46    | 88.06 | 80.94    | 87.72 |
> | 80                      | 75.92    | 83.82 | 72.35    | 82.84 | 79.03    | 85.22 | 78.95    | 85.59 |
> | $\beta$                 |          |       |          |       |          |       |          |       |
> | 0.1                     | 77.32    | 85.71 | 74.17    | 85.44 | 79.56    | 88.07 | 79.99    | 86.95 |
> | 1                       | 78.79    | 87.00 | 75.05    | 85.87 | 81.00    | 88.91 | 80.51    | 88.68 |
> | 10                      | 71.97    | 79.05 | 69.07    | 78.11 | 76.10    | 83.62 | 73.72    | 81.24 |
> | #Conv blocks/resolution |          |       |          |       |          |       |          |       |
> | 1                       | 78.79    | 87.00 | 75.05    | 85.87 | 81.00    | 88.91 | 80.51    | 88.68 |
> | 4                       | 79.81    | 88.08 | 74.80    | 85.28 | 81.56    | 88.67 | 80.59    | 87.19 |
> | 7                       | 78.725   | 86.80 | 73.48    | 84.70 | 79.29    | 86.69 | 79.30    | 86.59 |
> | #Augmented Samples      |          |       |          |       |          |       |          |       |
> | 2/chain                 | 74.78    | 80.09 | 69.23    | 79.33 | 75.70    | 84.63 | 73.90    | 84.01 |
> | 13/chain                | 78.79    | 87.00 | 75.05    | 85.87 | 81.00    | 88.91 | 80.51    | 88.68 |
> | 40/chain                | 71.32    | 79.12 | 66.09    | 77.28 | 74.09    | 81.20 | 72.05    | 82.28 |
> We make the following observations:
> - Under reasonable EBM hparams (smaller $\beta$, higher $k$), the performance of LangDAug is stable.
> - The performance remains stable with varying EBM complexity.
> - Smaller number of langevin samples is not effective as it doesn't capture the vicinal distributions.
> - Large number of langevin samples also harm performance due to high auto-correlation between langevin samples leading to biased learning. Hence, a well spaced selection of langevin samples is desired.
> ## Application to 3D Medical Images
> The primary challenge in extending LangDAug to 3D medical images is the current lack of mature methodologies for generating 3D volumes using EBMs. Current approaches [1] are limited to primitive 3D shapes and don't scale well to complex anatomical structures.
> As a partial evaluation, we applied LangDAug to the 3D Prostate MRI dataset by processing and augmenting 2D axial slices. While this doesn't capture full volumetric continuity, it provides evidence of LangDAug's effectiveness in 3D imaging contexts.
> We view full 3D EBM generation as an important direction for future work, which could be enabled by recent advances in score-based modeling and memory-efficient architectures.
>
> [1] Xie, Jianwen et al. "Learning descriptor networks for 3D shape synthesis and analysis." CVPR 2018.

---

### Official Review · Reviewer_G9sU · 2025-03-15

**Overall Recommendation:** 3

**Summary:**

This paper proposes a data augmentation method via Langevin dynamics with theoretical analyses. They also conduct experiments to show its usefulness.

**Claims And Evidence:**

In Line 019-023, "DA methods, which enrich model representations through synthetic samples, have shown comparable or superior performance to representation learning approaches." Is there empirical evidence for this?

**Essential References Not Discussed:**

N/A

**Experimental Designs Or Analyses:**

There seem to be too few baselines considering the abundant literature of domain generalization. To name a few: CORAL [1], RSC [2], SWAD [3], SagNet [4], etc.



[1] Sun B, Saenko K. Deep coral: Correlation alignment for deep domain adaptation[C]//Computer vision–ECCV 2016 workshops: Amsterdam, the Netherlands, October 8-10 and 15-16, 2016, proceedings, part III 14. Springer International Publishing, 2016: 443-450.

[2] Huang Z, Wang H, Xing E P, et al. Self-challenging improves cross-domain generalization[C]//Computer vision–ECCV 2020: 16th European conference, Glasgow, UK, August 23–28, 2020, proceedings, part II 16. Springer International Publishing, 2020: 124-140.

[3] Cha J, Chun S, Lee K, et al. Swad: Domain generalization by seeking flat minima[J]. Advances in Neural Information Processing Systems, 2021, 34: 22405-22418.

[4] Nam H, Lee H J, Park J, et al. Reducing domain gap via style-agnostic networks[J]. arXiv preprint arXiv:1910.11645, 2019, 2(7): 8.

**Methods And Evaluation Criteria:**

N/A

**Other Comments Or Suggestions:**

N/A

**Other Strengths And Weaknesses:**

In this paper, "DA" is short for data augmentation. However, in OOD literature, DA usually stands for domain adaptation. It is better to change another abbreviation for data augmentation to avoid confusion.

**Questions For Authors:**

What is the benefit of the proposed method in domain generalization specifically in medical image segmentation? It seems that it can be applied to normal domain generalization tasks as well.

**Relation To Broader Scientific Literature:**

N/A

**Theoretical Claims:**

N/A

---

> ### Author Rebuttal · Authors · 2025-03-30
>
> Thank you for your comments. We have tried our best to address your concerns by adding more baseline comparisons with your suggest methods. We will include these in the final version of the manuscript. We will be happy to answer any follow-up questions you might have.
>
> ## Comparison with more Baselines
> Thanks for this suggestion. The mentioned literature are slightly older DomainBed methods primarily designed for classification tasks. We have compared our method against methods like Fish, Fishr and Hutchinson which fall in same category. We provide the comparison of our method with the mentioned methods on RFS dataset (due to character limit) in the table below:
>
> | Baselines  | Domain 1 |       | Domain 2 |       | Domain 3 |       | Domain 4 |       |
> |------------|----------|-------|----------|-------|----------|-------|----------|-------|
> |            | mIoU     | mDSC  | mIoU     | mDSC  | mIoU     | mDSC  | mIoU     | mDSC  |
> | CORAL      |78.37     |86.55  |63.94     |74.20  |78.89     |87.71  |75.85     |85.65  |
> | RSC        |76.82     |86.11  |62.54     |72.42  |79.73     |87.92  |74.75     |85.04  |
> | SagNet     |75.42     |84.99  |54.51     |63.69  |75.52     |85.29  |67.31     |79.20  |
> | SWAD       |76.19     |84.99  | 65.32    |75.07  |76.91     |86.30  |72.09     |82.82  |
> | LangDAug       |**78.79**     |**87.00**  | **75.05**    |**85.87**  |**81.00**     |**88.91**  |**80.51**     |**88.68**  |
>
>
> We observe that our method outperforms these methods which is consistent to our observation with methods like Fish, Fishr and Hutchinson.
>
>
> ## Confusion for DA as Domain Adaptation
> Thanks for this suggestion. We will change this abbrevation from DA to DAug to avoid any such confusion.
>
> ## Benefit Specific to Medical Image Segmentation
> Thanks for this comment. While LangDAug can be potentially used for normal DG scenarios, we note that there are certain factors that guided our application design:
> - The effectiveness of LangDAug depends on the ability of EBMs to effectively traverse and interpolate between source domain distributions.
> - Such traversal becomes especially natural when source domains share structured similarities or consistent underlying factors of variation. Medical imaging data typically demonstrates structured variations, predominantly reflected through differences in amplitude spectra across domains [1,2].
> - EBMs have been shown to excel at capturing and modeling variations in amplitude spectra [3,4], making them suited for the domain variations characteristic of medical imaging data. Thus, LangDAug’s capability aligns well with the domain shift challenges encountered specifically in medical image segmentation.
>
> For these reasons, we use LangDAug in context of medical image segmentation.
>
> [1] Zhou et al. "Generalizable medical image segmentation via random amplitude mixup and domain-specific image restoration." ECCV, 2022.
>
> [2] Liu et al. "FedDG: Federated domain generalization on medical image segmentation via episodic learning in continuous frequency space." CVPR, 2021.
>
> [3] Du et al. "Compositional visual generation with energy based models." NeurIPS, 2020.
>
> [4] Tancik et al. "Fourier features let networks learn high frequency functions in low dimensional domains." NeurIPS, 2020.

---

### Decision · Program_Chairs · 2025-05-01

**Decision:**

Accept (poster)

**Comment:**

The paper proposes a Langevin Data Augmentation for multi-source domain generalization in 2D medical image segmentation. While the work presents an interesting approach, several critical concerns remain unresolved. During the rebuttal period, the concern that high computational requirement limits the application of the proposed method has not been adequately addressed. Besides, after reading the paper, reviews, and rebuttal, the submitted version misses some important content, like comparing it with other domain generalization methods, the ablation study of the proposed method, and broader experimental validation beyond fundus segmentation and 2D MRI datasets. Therefore, in my opinion, the current version does not meet the acceptance criteria for publication.